# Direct inversion of circulation from tracer measurements – Part 2: Sensitivity studies and model recovery tests

Thomas von Clarmann[1] and Udo Grabowski[1]

[1]Karlsruhe Institute of Technology, Institute of Meteorology and Climate Research, Karlsruhe, Germany

**Correspondence:** T. von Clarmann (thomas.clarmann@kit.edu)

**Abstract.** The direct inversion of the 2D continuity equation allows for the inference of the effective meridional transport of trace gases in the middle stratosphere. This method exploits the information given by both the displacement of patterns in measured trace gas distributions and the approximate balance between sinks and horizontal as well as vertical advection. Model recovery tests show that with the current setup of the algorithm, this method reliably reproduces the circulation patterns in the entire analysis domain from 6 to 66 km altitude. Due to the regularization of the inversion, velocities above about 30 km are more likely under- than overestimated. This is explained by the fact that the measured trace gas distributions at higher altitudes generally contain less information and that the regularization of the inversion pushes results towards zero. Weaker regularization would in some cases allow a more accurate recovery of the velocity fields but there is a price to pay in that the risk of convergence failure increases. No instance was found where the algorithm generated artificial patterns not present in the reference fields. Most information on effective velocities above 50 km is included in measurements of $CH_4$, CO, $H_2O$, and $N_2O$, while CFC-11, HCFC-22, and CFC-12 constrain the inversion most efficiently in the middle stratosphere. $H_2O$ is a particularly important tracer in the upper troposphere/lower stratosphere. $SF_6$ and $CCl_4$ contain generally less information but still contribute to the reduction of the estimated uncertainties. With these tests, the reliability of the method has been established.

## 1 Introduction

Traditionally, the observational analysis of the strength of the Brewer-Dobson circulation relies on the concept of the mean age of stratospheric air (AoA, Waugh and Hall, 2002). The AoA is the average transport time of an air parcel from the stratospheric entry point to the measurement location and is estimated from the mixing ratio of an age tracer such as $SF_6$. An alternative method, suggested by von Clarmann and Grabowski (2016, henceforth abbreviated vCG16), derives meridional circulation fields from two subsequent sets of global zonal mean vertically resolved pressure, temperature and mixing ratios of multiple long-lived trace gases by direct inversion of the continuity equation . This method is called "Analysis of the Circulation of the Stratosphere Using Spectroscopic Measurements" (ANCISTRUS). The resulting quantities are effective 2D velocities, that is to say, those 2D velocities which best describe the observed temporal changes of air density and constituent mixing ratio distributions by transport. They thus include all effects caused by longitudinal or temporal correlations between mixing ratios

and velocities. The relationship of these effective 2D velocities to 3D velocities is discussed in the Appendices of vCG16 and von Clarmann et al. (2019, henceforth vC19). Beyond this, the ANCISTRUS-derived effective velocities currently include also a contribution by physical mixing and thus are not directly comparable to the 2-D residual circulation in the Transformed Eulerian Mean framework.

Similar as in other applications of inverse modelling, such as retrieval of atmospheric state variables from radiance measure-
ments (e.g., Rodgers, 2000) or data assimilation (e.g., Ide et al., 1997), each iteration of the inversion scheme in ANCISTRUS consists of two steps: A forward modelling step and the inversion itself. In the forward modelling step, the current guess of the effective velocity field is applied to an initial field of measured atmospheric state variables (air density and mixing ratios of species) to solve the predictive version of the continuity equation. Sinks of trace gases due to photolysis, OH-chemistry and $O^1D$ chemistry are considered as described in vC19. Along with this, the partial derivatives of each atmospheric state variable
with respect to each element of the velocity vector are calculated. In the inverse step, the predicted field of the atmospheric state variables is compared with its measured counterpart, and the weighted residual is minimized by inverting the continuity equation. The weights are represented by the inverse covariance matrix, including measurement uncertainties and prediction errors. To keep the inversion stable, a constraint is applied.

The natural application of this method is the analysis of the Brewer-Dobson circulation (Brewer, 1949; Dobson, 1956).
ANCISTRUS avoids certain drawbacks of the hitherto common method using the mean age of stratospheric air (Waugh and Hall, 2002) as a diagnostic of the circulation. No age spectra (Andrews et al., 1999; Waugh and Hall, 2002) have to be assumed. Intrusion of mesospheric $SF_6$-depleted air does not cause artificial "overaging" of the air (Stiller et al., 2012; Reddmann et al., 2001; Ray et al., 2017), because for gases without a stratospheric sink, ANCISTRUS takes all information from mixing ratio differences within the analysis domain and not from the absolute abundances. Age-of-air based methods exploit the measured
mixing ratio difference between the stratospheric entry point and the measurement location, and the air might have been depleted in $SF_6$ during its potential detour through the mesosphere. The mesospheric loss of $SF_6$ increases the difference and makes the air appear older than it actually is. In contrast, ANCISTRUS exploits the measured difference in the mixing ratios of $SF_6$ between the endpoint and the starting point of a path element of the trajectory only in the domain considered. If the air parcel has re-entered the analysis domain after a possible detour through the mesosphere, any mesospheric loss has affected
both the starting point and the endpoint of the path element and thus does not contribute to the difference. And finally, the method does not provide the integrated travel time of an air parcel only but provides time-resolved results.

Applying ANCISTRUS to trace gas mixing ratios measured with the Michelson Interferometer for Passive Atmospheric Sounding (MIPAS, Fischer et al., 2008) results in circulation fields that include the expected features like tropical uplift, polar winter subsidence, stratospheric poleward transport, mesospheric pole-to-pole circulation, and elevated stratopauses (vC19).
Furthermore, results proved to be stable in the sense that for each year – within the expected range of variability – similar circulation fields were found for any particular time of the year, although the estimates were independent from each other. The ANCISTRUS version used in this paper includes several upates with respect to the original method by vCG16. In particular, sinks of trace gases are considered and mixing coefficients are constrained to zero. The latter implies that resulting velocities are effective velocities that also account for the effect of eddy mixing and physical diffusion. Further details are reported

in vC19. Application to trace gas distributions obtained from other satellite missions, such as the Microwave Limb Sounder (MLS, Waters et al. 2006) or the Atmospheric Chemistry Experiment – Fourier Transform Spectrometer (ACE-FTS, Bernath et al. 2005) is under consideration.

Since chemical decomposition has been newly implemented in the most recent ANCISTRUS version, the effect of the consideration of sinks is investigated in Section 2. The purpose of this investigation is to find out how much information on the circulation is provided by the sinks and how much is provided by the displacement of mixing ratio patterns. In order to further increase the confidence in the new inversion-based method, in this paper we validate the inverse method by model recovery tests. For these tests, mixing ratio distributions are modeled using known effective velocities. These mixing ratio distributions are then fed into ANCISTRUS to test how well the initial velocity field is recovered (Section 3).

These tests are complemented by an assessment of the dependence of the results on the regularization strength (Section 4). Further, we study the sensitivity of the model to the availability of various trace gas fields (Section 5). In the Conclusions (Section 6) we discuss the power and the limitations of the method as discovered in this work, and make suggestions for further work.

## 2   Sinks versus transported structures

Two mechanisms link mixing ratio distributions with the circulation and thus allow to retrieve information on the circulation from measured mixing ratio distributions. One mechanism is the interplay between the chemical destruction of trace gases and advection. Without advection, chemical sinks would remove those gases which have their sources at Earth's surface completely from the stratosphere, and the fact that we observe – in the long run, and putting weak long-term trends aside – approximately stationary trace gas distributions can only be explained by horizontal and/or vertical advection. Roughly speaking, with the assumption of a chemically stationary atmosphere in force, i.e., when mixing ratio distributions are assumed not to change with time, at each point of the atmosphere the loss by chemical decomposition is compensated by advection of the related species. That is to say, if a molecule is destroyed, another molecule of this species must be brought to this point by transport if the stationarity condition shall be satisfied. This defines a circulation field corresponding to an equilibrium with respect to atmospheric composition. Mixing ratios changing with time can be understood as a perturbation of this equilibrium assumption, but the task could be conceived as finding the equilibrium circulation where transport balances decomposition. Needless to say that this requires the modelling of sinks in the forward model that is used to predict the atmospheric state. In the current version of ANCISTRUS, the sinks of $CCl_4$, CFC-11, CFC-12, $CH_4$, CO, HCFC-22, $H_2O$ and $N_2O$ are considered as described in vC19, while, due to its long stratospheric lifetime, $SF_6$ is considered as inert in the given analysis range. For CO and $H_2O$ also source reactions are considered. For reasons discussed above, ANCISTRUS is sensitive only to decomposition of gases within the diagnosed latitude and altitude range but not to depletion at higher altitudes. Any depletion of, say, $SF_6$ on its way through the mesosphere before it subsides again into the stratosphere thus does not affect the ANCISTRUS results.

The other mechanism by which trace gas distributions convey information on the circulation is the transport of structures. If, say, the maximum of the mixing ratio of a certain gas is at a certain location one day, and 5 degrees further south a month

later, this is best explained by a southward velocity of 5 degrees per month, assuming that this solution satisfies the continuity equation globally. The amplitude of the structures transported is affected by the sinks discussed above. A widely used method that uses this information pathway is the analysis of the ascent rate in the tropical pipe by means of the water vapour tape recorder (Mote et al., 1996).

As opposed to both these simplified views where information pathways are assessed in isolation, both mechanisms contribute to the full picture. ANCISTRUS thus exploits both information pathways. In order to test the sensitivity of ANCISTRUS with respect to each of them, the following tests were performed: As a reference, we use a regular ANCISTRUS result based on zonal mean MIPAS measurements of all 9 trace gases from March to April 2005 (Fig. 1, upper left panel) and for September to October 2010 (Fig. 2, upper left panel). The choice of these years has no particular reason; the seasonal behaviour of these years is well representative for that of the other years available. The months March–April and September–October were chosen because the velocity fields are more structured than at other times of the year and thus more interesting for test purposes.

The circulation fields roughly match our expectations of a typical middle atmospheric meridional circulation. We see mesospheric/upper stratospheric subsidence in local autumn. The mesospheric pole-to-pole circulation is more pronounced in September-October 2010 than in March-April 2005. Poleward transport in the lower and middle stratosphere is associated with the Brewer-Dobson-circulation. Northern polar upwelling in March-April 2005 is particularly interesting: this is explained by the displacement of the polar vortex off the pole during the sudden stratospheric warming taking place at this time, which means that at the pole strongly subsided vortex air is replaced by less subsided air, resulting in a local (Eulerian) upwelling in a 2D perspective. Due to symmetry around the pole, in a 2D representation there is no horizontal velocity which could reproduce this phenomenon. This result, seeming counter-intuitive at first glance, is not a weakness of the ANCISTRUS method but rather a characteristic of the representation of the 3D atmosphere in 2D in general.

While the scientific interpretation of these fields of effective velocity is provided elsewhere (e.g., vC19), we are, within the framework of this technical study, not so much interested in the explanation of the atmospheric features but in the sensitivity of the inversion with respect to changes in the setup. The upper right panels of Figs. 1 and 2 show the respective ANCISTRUS run without the consideration of chemical sinks. The structures and circulation patterns described before are still present, but the velocities have changed in a quantitative sense. An additional feature of equatorward transport at about 55 km altitude, 30°S has emerged in March April 2005. As expected, the relevance of sinks is largest at higher altitudes but in general it is moderate in a sense that minor inaccuracies in sink strengths are not likely to perturb the general picture of the circulation.

By feeding ANCISTRUS with identical trace gas fields for the beginning and the end of the time interval under consideration, the equilibrium circulation was inferred, where sinks are completely balanced by advection (bottom panels). We have performed two variants of this test.

In the first variant, ANCISTRUS was fed with the actual trace gas measurements for the first month, and with the same distribution for the second month. The goal was to emulate steady state conditions and to remove all information contained in the transport of mixing ratio patterns. Here the general picture changes dramatically. Several features of the reference case are not seen anymore. These include the strong subsidence over the South pole, the response to the stratospheric warming over the North pole, and poleward transport below 20 km in both hemispheres, in March–April 2005 (Fig. 1, lower left panel).

The tropical upwelling reaches up into the mesosphere. For September–October 2010 the pole to pole circulation is no longer present (Fig. 2, lower left panel). Two fairly symmetric circulation cells with maximum poleward effective velocities in 50–60 km dominate the velocity field. Again, the tropical upwelling reaches up into the mesosphere.

Since, strictly speaking, monthly mean mixing ratios do not represent a genuine steady state but rather a snapshot of a transient state, we have repreated this test using annual mean mixing ratio distributions. Without information on monthly changes of the atmospheric state, and no seasonal information in the mixing ratio distribution, the inferred circulation is fairly symmetrical, regardless of sinks being estimated with lifetimes typical for March/April (Fig. 1, lower right panel) or September/October (Fig. 2, lower right). With this setup, the tropical upwelling again reaches up into the mesosphere, and the remaining patterns are two rather symmetric transport cells in each hemisphere, the stronger one around 50 km covering all hemispheric latitudes, and a weaker one around 25 km, located in the subtropics. In summary, it is evident that both sources of information contribute to the resulting circulation field, and it is necessary to exploit both of them to infer a realistic circulation field.

## 3  Model recovery tests

vCG16 have presented two series of tests. In a first step, they tested the implementation of the transport scheme used. Tests were chosen intentionally simple in order to make it possible to judge if the algorithm does what it is supposed to, without involving the need of a separate model. If a structure, e.g., a mixing ratio maximum, is transported northward by 5 degrees in one month when the assumed uniform velocity field is 5 degrees per month, the success of the test can be directly judged. Diffusive and dispersive characteristics can be tested by analysis of the size of the transported maximum and side wiggles created during the transport. Neither indication of any malfunction nor otherwise conspicious features were found in a long series of these forward model tests of which a small subset was shown in vCG16. This kind of test is considered as severe in the sense of Mayo (1996) because the probability that a flawed transport scheme would be detected is large. Thus, the likelihood that a model which passes these tests is flawed is small. Despite their simplicity, these tests are also general because the operations of the transport scheme are the same everywhere in the analysis space. We thus consider the transport scheme used by ANCISTRUS as valid.

vCG16's second series of tests focused on the inversion scheme. Tests fully based on trace gas real measurements suffer from the fact that the corresponding true velocity fields are not known and it is thus not clear what the resulting effective velocity fields should be compared to. Model recovery tests based on assumed velocity fields used as surrogate truth along with simulated measurements avoid this problem. Such a test is organized as follows. The assumed velocity field is taken as a reference field and is applied to a measured initial atmospheric state. The resulting solution of the forward transport problem renders the simulated state at a later time. Then the measured initial and the simulated later atmospheric state are fed into the inversion scheme as surrogate measurements, and the resulting velocity field, recovered without using any information on the surrogate truth, is compared to that reference field used to simulate the later atmospheric state.

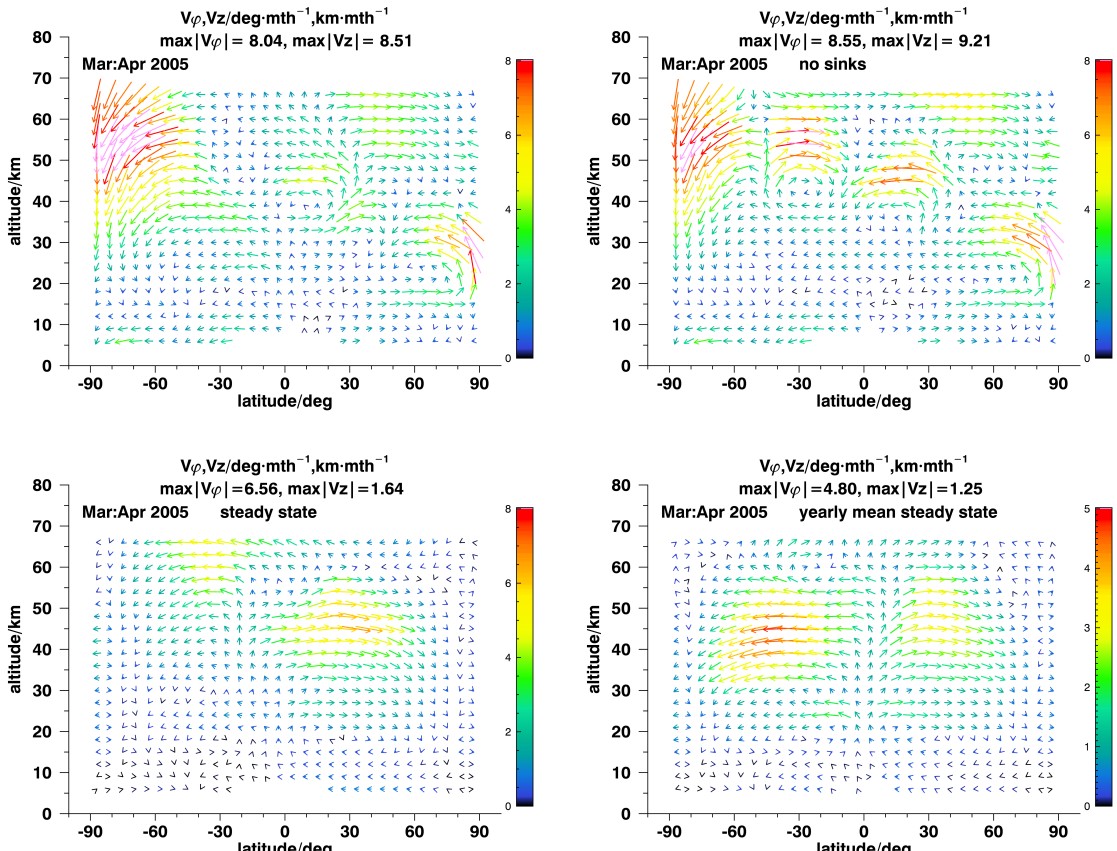

**Figure 1.** : The meridional middle atmospheric circulation as retrieved with ANCISTRUS for March-April 2005 under realistic assumptions (upper left panel), without consideration of sinks of trace gases (upper right panel), and for sinks perfectly balanced by transport for actual (lower left panel) and annual mean (lower right panel) conditions. The colour scales refer to $\sqrt{(v_\phi \mathrm{deg}^{-1}\mathrm{month})^2 + (v_z \mathrm{km}^{-1}\mathrm{month})^2}$ for $v_\phi$ and $v_z$ in units of deg month$^{-1}$ and km month$^{-1}$. Pink arrows refer to velocities higher than representable by the colour scale chosen.

160      For these tests, a sensible choice of the assumed velocity field is essential. Related tests by vCG16 were based on an *ad hoc* choice of the velocity field. Again, the broad functionality of the inversion scheme could be demonstrated but a closer look revealed that these tests were only partially successful. The cause of problems encountered was that the velocity fields used for testing were not solutions of the continuity equation. An inversion scheme that is based on the hard-wired constraint that the results must comply with continuity cannot reproduce velocity fields which were chosen in an *ad hoc* manner and are not

165 compliant with continuity. Thus, spurious test results at the boundaries of the analysis field did not come unexpected and could not refute the validity of the algorithm.

     More severe tests thus must use a velocity field that satisfies the continuity equation. On the face of it, tracer and velocity fields from a chemistry-climate model or a chemistry-transport-model would serve the purpose. The comparison of ANCISTRUS results with those from such a model, however, suffers from the fact that 2D velocities cannot be unambigously

170 compared to 3D model results because there is some room for interpretation of the 2D effective velocities. The latter include

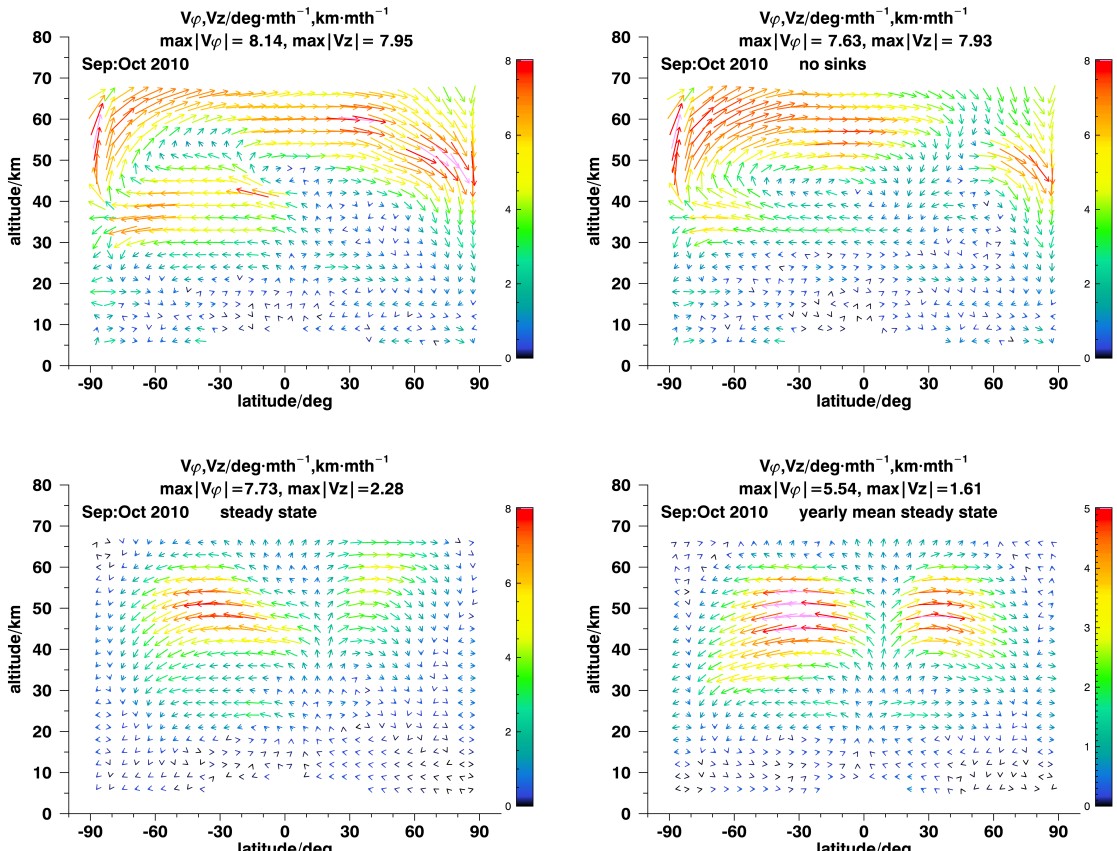

**Figure 2.** : The meridional middle atmospheric circulation as retrieved with ANCISTRUS for September-October 2010 under realistic assumptions (upper left panel), without consideration of sinks of trace gases (upper right panel), and for sinks perfectly balanced by transport for actual (lower left panel) and annual mean (lower right panel) conditions. For details, see Fig. 1

contributions from eddy transport and eddy mixing (See appendices in vCG16 and vC19). Furthermore, there exist some more technical problems: Often the zonal mean mixing ratio fields from the climate model deviate in a sizeable way from the MIPAS profile. In this case it is not clear what uncertainties shall be assigned to these mixing ratios from the model. Any rescaling of the assumed error variances would substantially change the weights of the measurements in the inversion, and the results
175 would no longer be representative for the application of ANCISTRUS to MIPAS zonal means. Beyond this, modelled trace gas fields are often less structured than the measured ones. The absence of prominent structures, however, means the absence of some useful information for ANCISTRUS, again leading to results not directly comparable to the application of ANCISTRUS measurements to MIPAS trace gas fields.

The use of velocities from a model applied to MIPAS volume mixing ratios to generate mixing ratio fields at the second time
180 step does not solve the problem either. The reason is this. As we have learned from the tests in Section 2, the velocities and the initial mixing ratio distributions cannot be chosen independently. For species with sinks in the stratosphere, not only the mixing ratio differences between the beginning and the end of a time step depend on the velocities, but also the absolute concentrations

and their spatial distributions. Inconsistencies between the velocity field and the mixing ratio distributions would thus lead to artefacts in the result of the test. A test where it is not possible to decide if any discrepancy between the reference velocity field and the retrieved velocity field is due to this type of artefact or to a possible malfunction of ANCISTRUS is not useful for validation purposes.

Our way out is to use ANCISTRUS-generated effective velocity fields to simulate trace gas and density fields, apply AN-CISTRUS to them, and test the resulting velocity field by comparison to the initial velocity field. The ANCISTRUS-generated effective velocity fields satisfy the continuity equation. One might argue that this type of model recovery test is circular, but the circularity is related only to the forward transport model which has already been tested independently. Further, this test of the inversion scheme takes fully place in a two-dimensional world and thus avoids any complication by the interpretation of 2D effective velocities and their relation to 3D model results.

Results of our model recovery tests are shown in Fig. 3 for March–April 2005 (left panels) and for February–March 2010 (right panels) and in Fig. 4 for August–September, 2010 (left panels) and September–October 2010 (right panels). Figures 5 and 6 with their reduced altitude range permit a closer look at the lower stratosphere. Top panels show the reference fields of effective velocity, middle panels show the recovered fields, and the respective differences are shown in the bottom panels. The usual diagnostics were applied and in none of the cases any peculiarities were detected. This provides evidence that the system of equations solved has an unambiguous solution.

For the March–April 2005 case (Figs. 3 and 5, left panels), ANCISTRUS reproduces all the patterns of the reference case: subsidence of mesospheric air into the stratosphere at Antarctic latitudes, stratospheric effective upwelling over the North Pole, the bifurcation of an upwelling circulation segment at 30°N, 45 km altitude, poleward transport in the southern hemispheric subtropics at 25 km altitude and in the northern hemispheric subpolar region at 15 km altitude. All these features are recovered at the correct altitudes and latitudes. At Antarctic latitudes around 55 km altitude effective velocities are under-estimated by 8–9% while they are over-estimated at 40°S, 40 km, by up to 20%. The center of the circulation structure at tropical latitudes at around 45 km is shifted downward by 3 km. Largest relative deviations are found where the reference case contains circulation segments in opposite directions at adjacent altitudes. The Tikhonov regularization chosen is designed to keep velocity differences between adjacent model gridpoints small. Thus, this kind of smoothing error observed where the inversion cannot fully resolve the reference field does not come unexpected. Also the structures of the slow circulation patterns in the tropopause region and the lower stratosphere are well recovered (Fig 5, left panels). Effective poleward velocities at 6 and 9 km altitude and the northward effective velocities in northern midlatitudes at 15 and 18 km are underestimated in some places.

For the February–March 2010 test case, the situation is very similar to the one discussed above (Figs. 3 and 5, right panels). Again, we see southern polar subsidence and the bifurcation of the upwelling circulation segment at 30°N, 45 km altitude. Contrary to March–April 2005, we see subsidence also over the North Pole, which is an expected phenomenon in polar winter vortices. All major circulation patterns are recovered at the correct latitudes and altitudes. Peak velocities in the mesospheric branches of the circulation are underestimated by about 25% but in large parts of the analysis domain the inversion is successful also in quantitative terms, particularly below 40 km. Again, largest discrepancies are found where opposite circulation directions are found at adjacent gridpoints: Due to the smoothing regularization, the inversion does not resolve the small circulation

feature at 20°S, 45 km altitude. A more detailed view on the lower altitudes (Fig 5, right panels) shows that the branches of the Brewer-Dobson circulation are well recovered (20°S–40°S at 21–27 km altitude and in northern midlatitudes at altitudes between 18 and 27 km). The latitudes, altitudes, and velocity values of maximum poleward transport agree well. Also the position, altitude and strength of tropical upwelling is almost perfectly recovered.

Tests for August–September 2010 and September–October 2010 (Figs 4 and 6) confirm the findings of the first two tests. All patterns and structures are recovered at the correct latitudes and altitudes. For August–September 2010, this refers to the bifurcation of upward and downward effective velocities in the southern polar upper stratosphere near 40 km; the huge area of large southward velocities at 33–60 km altitude between Equatorial latitudes and about 70°S; the local maxima of southward effective velocities at 24–27 km at about 10°S and at 6–9 km at southern midlatitudes; the position of the upwelling within the tropical pipe around 10°N, a large area of high northward velocities peaking between 54 and 60 km in northern midlatitudes and feeding into northern polar subsidence. Peak velocities are underestimated by about 20%. Quantitative deviations between the reconstructed field and the reference field are largest where velocity gradients are largest. E.g., the bifurcation of tropical upwelling velocities between 40 and 50 km is not well resolved, due to the smoothing characteristic of the regularization.

The September-October 2010 circulation (Figs. 4 and 6, right panels) is characterized by a strong northward mesospheric pole-to-pole circulation which is connected to southward transport between 30 and 50 km in the entire southern hemisphere. The general structure of this circulation system and the positions of peak velocities are almost perfectly recovered but peak velocities are underestimated by about 20%, again due to the smoothing regularization. Poleward velocities at 6 and 9 km in southern midlatitudes, 21–27 km between 20°N and 60°N as well as equatorward velocities at 15 km altitude in midlatitudinal and polar northern latitudes are all recovered.

Most importantly, in none of the tests, the inversion scheme has created artificial patterns which were not present in the reference case. No major pattern was removed. The small-scale circulation feature at 20°S, 45 km altitude in February–March 2010 (Fig. 3, right panels) is the only instance of a feature in the reference field which has not been reproduced.

## 4 The role of the regularization strength

In the previous section, the fact that large velocities are not fully recovered is attributed to the regularization of the inversion. ANCISTRUS uses a Tikhonov (1963) type regularization which leads to the following object function to be minimized:

$$(\boldsymbol{x} - \boldsymbol{F}(\boldsymbol{q}; \boldsymbol{x}_0))^T \mathbf{S}_r^{-1} (\boldsymbol{x} - \boldsymbol{F}(\boldsymbol{q}; \boldsymbol{x}_0)) + \boldsymbol{q}^T \mathbf{L}_1^T \boldsymbol{\Gamma} \mathbf{L}_1 \boldsymbol{q} \tag{1}$$

$(\boldsymbol{x} - \boldsymbol{F}(\boldsymbol{q}; \boldsymbol{x}_0))$ is the residual between the measured field $\boldsymbol{x}$ of atmospheric state variables and those predicted using the initial field $\boldsymbol{x}_0$ and an assumed field of velocities $\boldsymbol{q}$. All these fields are expressed as vectors of length $m$. $\mathbf{S}_r$ is the $m \times m$ covariance matrix characterizing the uncertainties of the residual, under consideration of uncertainties of $\boldsymbol{x}$ and $\boldsymbol{x}_0$. $\mathbf{L}_1^T \boldsymbol{\Gamma} \mathbf{L}_1$ is the $n \times n$ regularization term, where $\mathbf{L}_1$ is a first order difference matrix of dimension $(n-1 \times n)$, expressing the vertical and horizontal differences of adjacent values of horizontal and vertical velocities. These velocities are represented by the $n$-dimensional vector $\boldsymbol{q}$. $\boldsymbol{\Gamma}$ is a diagonal $(n-1) \times (n-1)$ matrix and controls the strength of the regularization and balances the

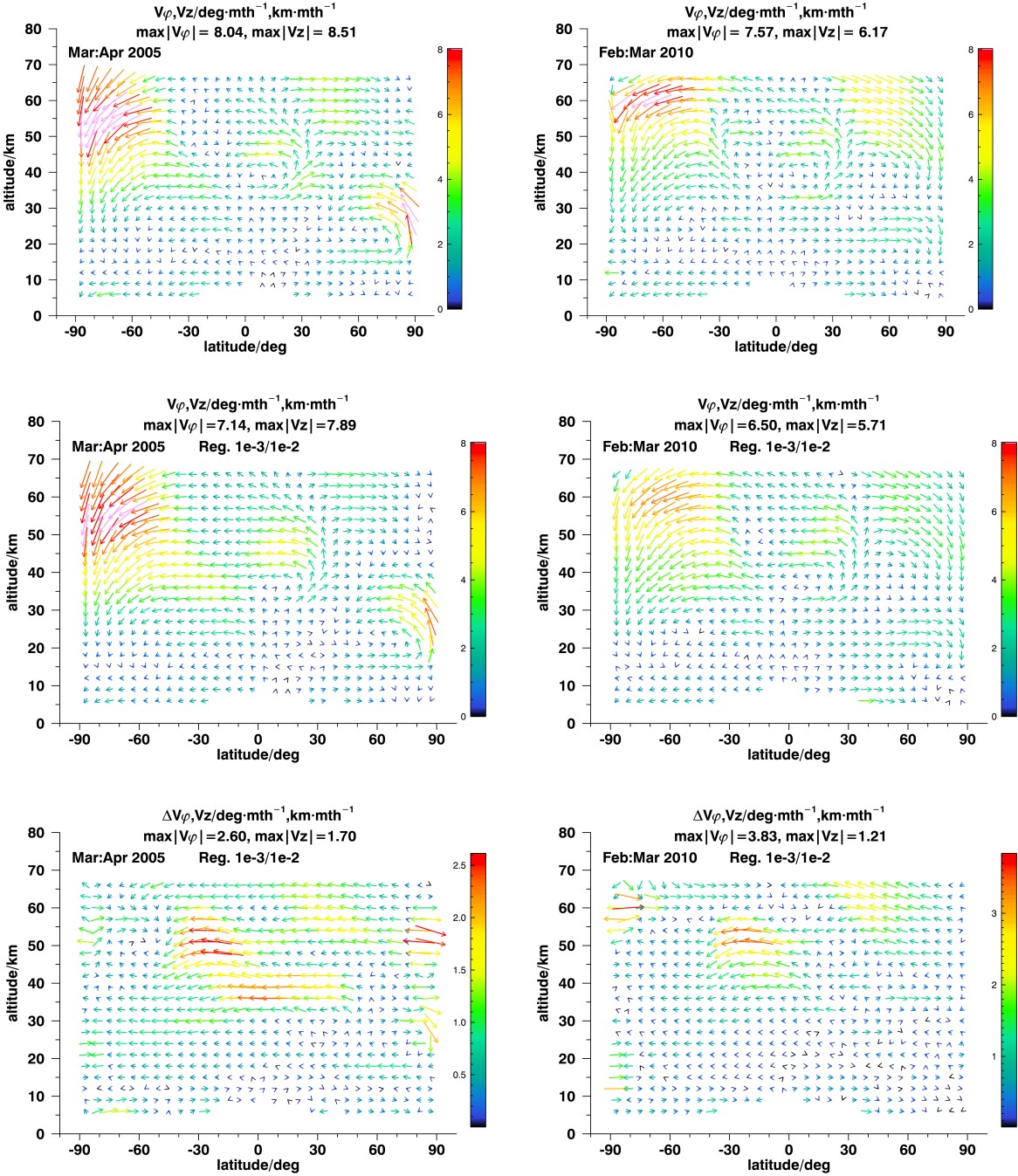

**Figure 3.** : Model recovery tests for March–April, 2005 (left panels) and February March, 2010 (right panels), reference fields (top panels), results (middle row) and differences (bottom panels). Note the different colour scales of the difference plots. For details, see Fig. 1.

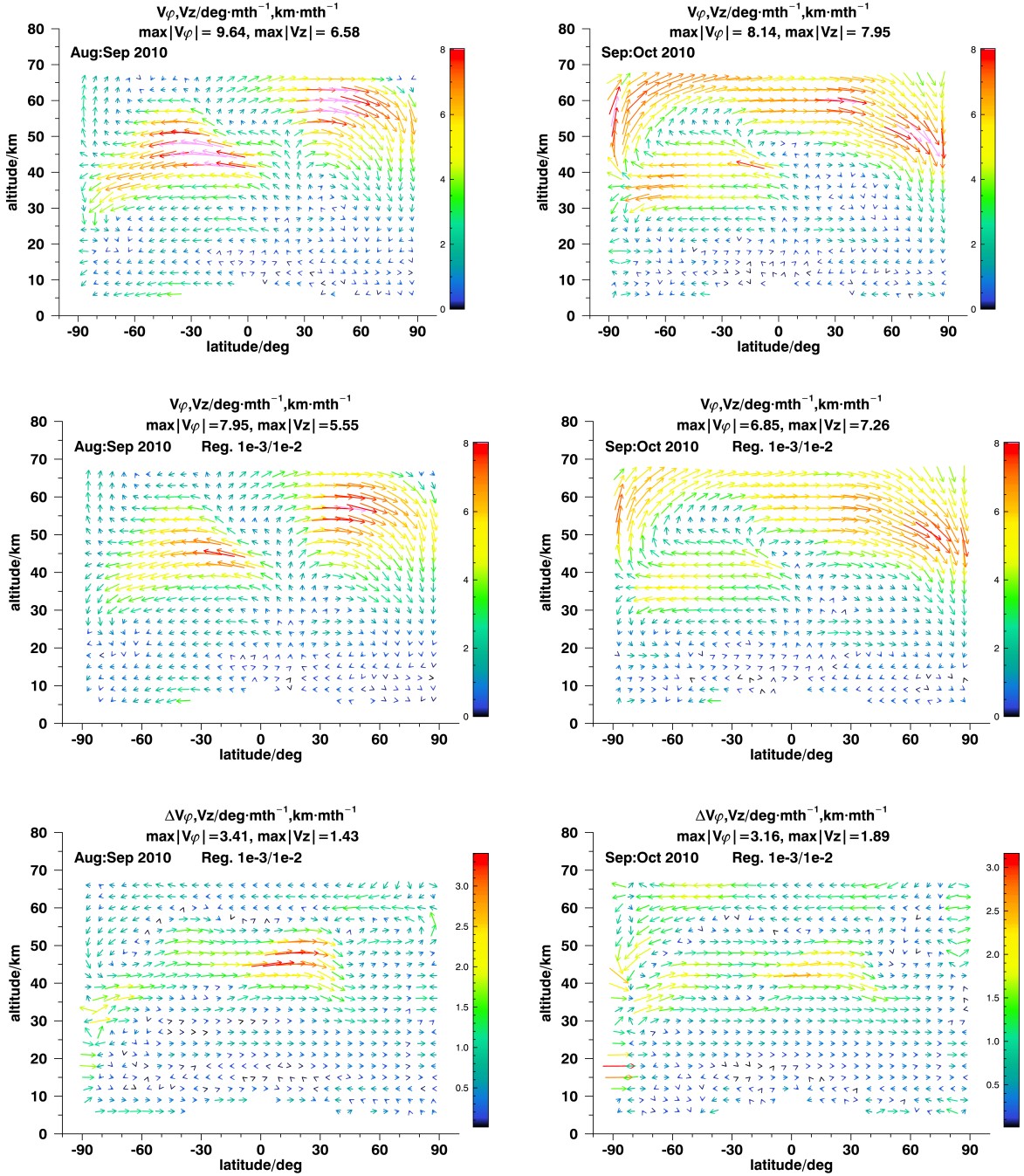

**Figure 4.** : Model recovery tests for August–September, 2010 (left panels) and September–October 2010 (right panels). For details, see Fig. 3.

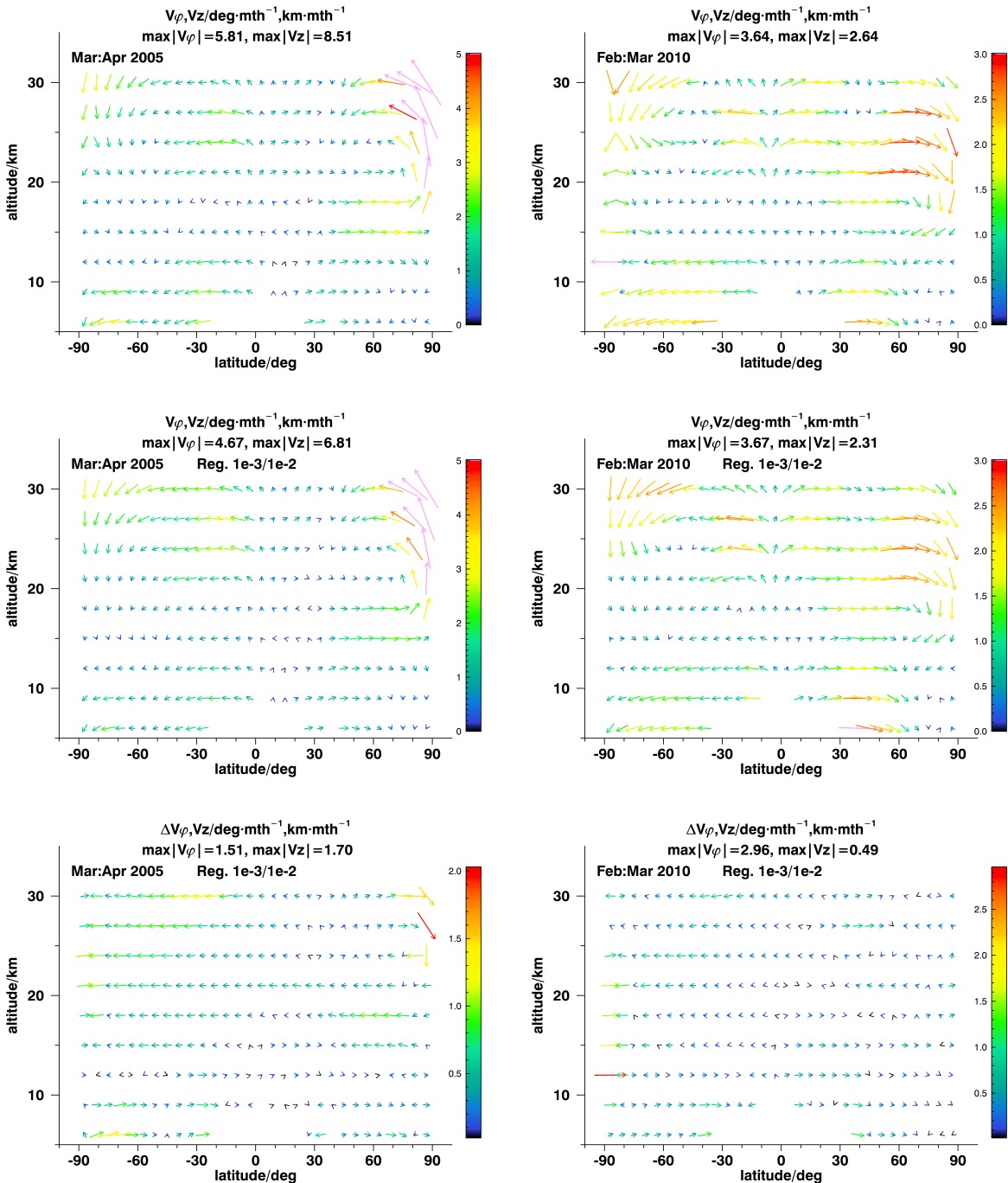

**Figure 5.** : As for Fig. 3 but with a reduced altitude range for clearer representation of lower altitudes.

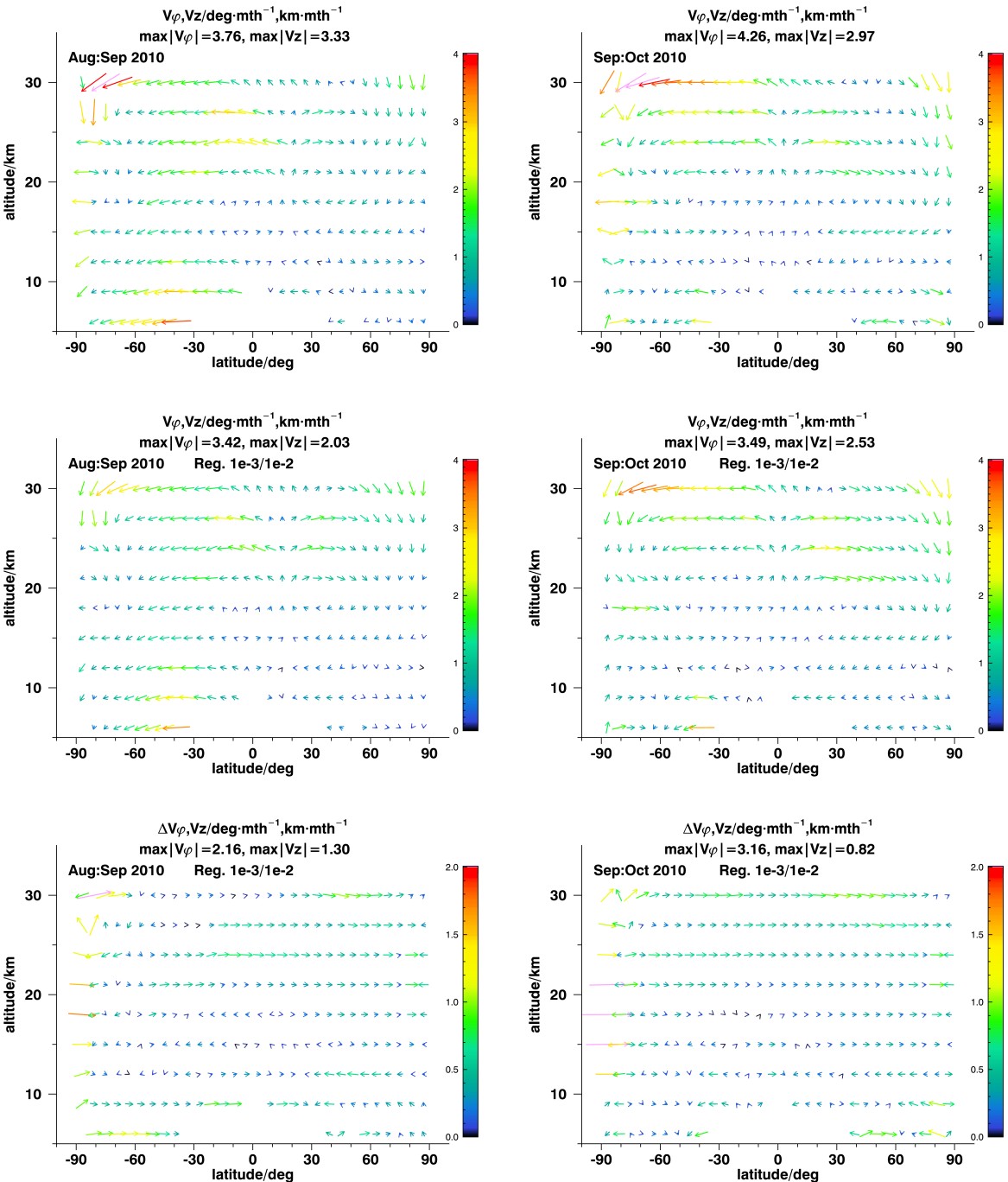

**Figure 6.** : As for Fig. 4 but with a reduced altitude range for clearer representation of lower altitudes.

units. The purpose of the regularization term is to prevent horizontal or vertical gradients of horizontal and vertical velocities from becoming unreasonably large, a typical characteristic of instable, oscillating solutions of ill-posed inverse problems. It goes without saying that the choice of the entries of $\Gamma$ directly affects the solution. Thus it is in order to test how sensitive the resulting velocity fields are on the choice of $\Gamma$. We use September-October 2010 as a test case, because the large velocity contrasts are a particular challenge for a Tikhonov-type smoothing regularization.

For September–October 2010, the model recovery test presented in the previous section relied on regularization strengths of $(c_1 \times 1.0 \times 10^{-3})^2$ for all entries of $\Gamma$ operating on horizontal velocities and $(c_2 \times 1.0 \times 10^{-2})^2$ for those operating on vertical velocities. $c_1$ and $c_2$ were $6.0 \times 10^4$ m$^{-1}$s and $1.0 \times 10^6$ m$^{-1}$s, respectively. In addition, the following pairs of regularization strengths were tested: $((c_1 \times 5 \times 10^{-3})^2; (c_2 \times 5.0 \times 10^{-2})^2)$, $((c_1 \times 1 \times 10^{-2})^2; (c_2 \times 1.0 \times 10^{-1})^2)$, $((c_1 \times 5 \times 10^{-4})^2; (c_2 \times 5.0 \times 10^{-3})^2)$ and $((c_1 \times 3 \times 10^{-4})^2; (c_2 \times 3.0 \times 10^{-3})^2)$. Results are presented in Fig. 7.

For the two strongest regularizations the main circulation is qualitatively reproduced but velocities are underestimated by a factor of two to three. Details of the field are not well resolved (lower and middle left panels). With regularization strengths $(c_1 \times 1.0 \times 10^{-3}; c_2 \times 1.0 \times 10^{-2})$, which is the one usually applied, all patterns are well resolved, and approximate quantitative agreement is found almost everywhere, except for the peak velocities, which are underestimated by several ten percent (upper right panel). With regularization strengths of $(c_1 \times 5.0 \times 10^{-4}; c_2 \times 5 \times 10^{-3})$ the agreement is even better, but there are a

significant number of cases for other months where no convergence of the iterative inversion could be obtained (middle right panel). An even weaker regularization of $(c_1 \times 3.0 \times 10^{-4}; c_2 \times 3.0 \times 10^{-3})$ gives room to some instabilities at the boundaries of the domain, particularly at the South Pole between 15 and 30 km altitude (bottom right panel). Thus, we consider the nominal regularization strengths as adequate for routine processing. The damping of peak velocities is the price to pay for a robust inversion. With rare cases of non-convergence a good data coverage can be achieved, structures and patterns can safely be

recovered, and outside the regions of peak velocities the results are robust even in a quantitative sense. The optimal choice of the regularization strength, however, is application-dependent, and for particular case studies, where convergence turns out not to be a problem, a weaker regularization may be more adequate.

## 5 Sensitivity tests

For several reasons, ANCISTRUS results are expected to depend on the selection of species used. First, species with different

concentration profiles carry information on the circulation at different altitudes. Thus, omitting, e.g., CO and CH$_4$ and using only species with sizeable concentrations in the lower stratosphere, like CCl$_4$ or CFC-11, will lead to heavily degraded results in the mesosphere. Second, the more species we have in general, the weaker the effect of regularization will be and thus more information can be retrieved, even if the additional information does not change the result in any appreciable manner. Thus, the sensitivity of results with respect to the omission of single species is worthwhile testing. A low sensitivity to the omission

of a single species shows the robustness of the methodology.

The corresponding test was set up as follows: First, an ANCISTRUS run was performed for a complete set of species. Then, a series of ANCISTRUS runs was performed, each with one gas omitted, similar as to a jackknife method. The difference of

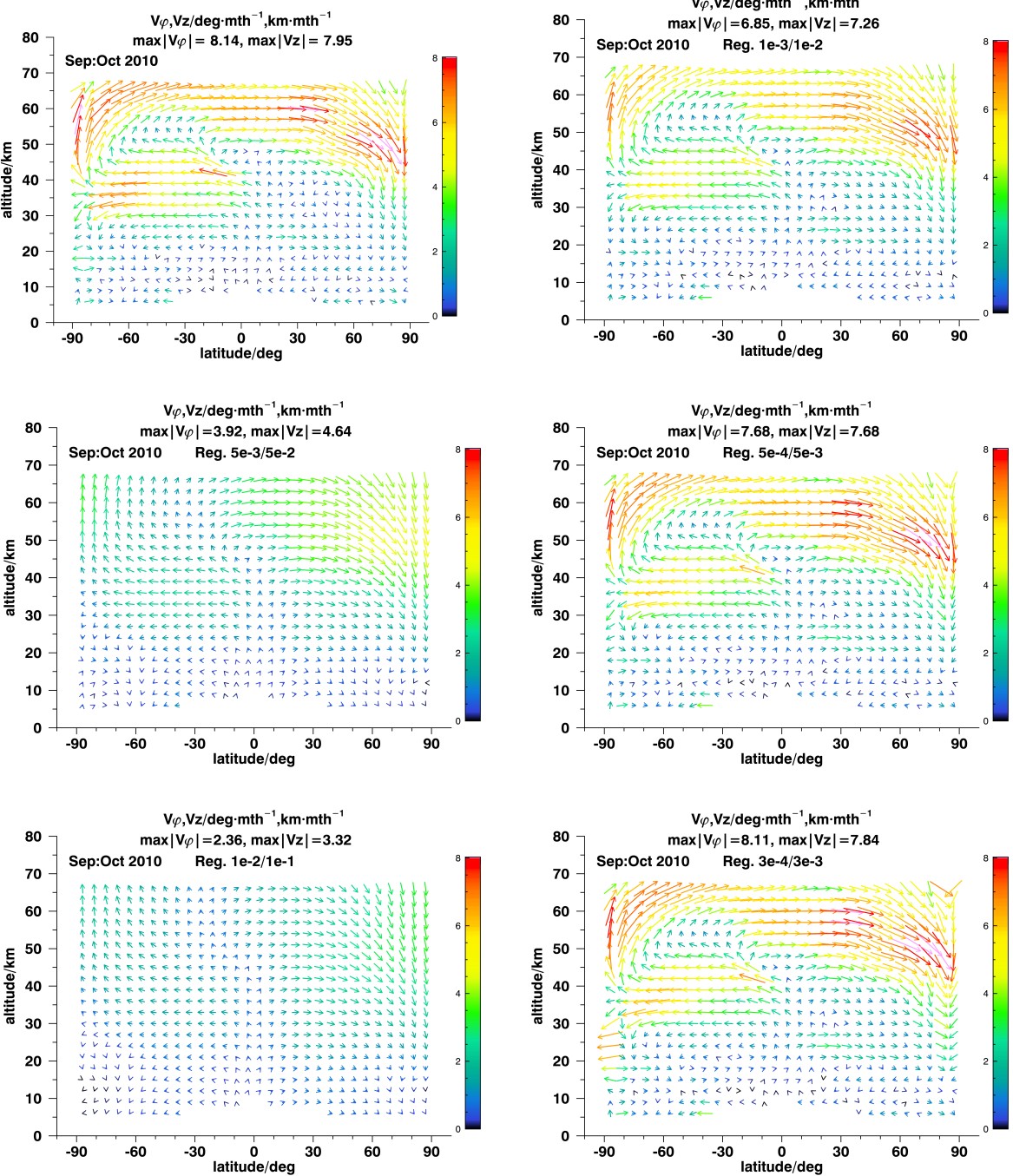

**Figure 7.** Resulting fields of effective velocity for different regularization strengths. The upper left panel shows the reference velocity distribution and the upper right panel the model recovery test for the nominal regularization strength of $(c_1 \times 1.0 \times 10^{-3})^2$ for horizontal velocities and $(c_2 \times 1.0 \times 10^{-2})^2$ for vertical velocities. The left middle and lower panels show results for stronger regularization of $((c_1 \times 5.0 \times 10^{-3})^2; (c_2 \times 5.0 \times 10^{-2})^2)$ and $((c_1 \times 1.0 \times 10^{-2})^2; (c_2 \times 1.0 \times 10^{-1})^2)$, respectively. The right middle and lower panels show results for weaker regularization of $((c_1 \times 5.0 \times 10^{-4})^2; (c_2 \times 5.0 \times 10^{-3})^2)$ and $((c_1 \times 3.0 \times 10^{-4})^2; (c_2 \times 3.0 \times 10^{-3})^2)$, respectively.

velocities caused by the omission of a candidate species is a measure of the sensitivity of the retrieval to this species. These tests were performed for March–April 2005 (left panels) and September–October 2010 (right panels) for the omission of CFC-11,

CFC-12 and HCFC-22 (Fig. 8), $CCl_4$, $SF_6$, and $H_2O$ (Fig. 9), as well as $N_2O$, $CH_4$, and CO (Fig. 10). CFC-11, CFC-12, and HCFC-22 contribute most in the polar spring stratosphere, where gradients between regions of old air depleted in these species and young air rich in these species are large (Fig. 8). Since mixing ratios of these species are low in the upper stratosphere and above, these species contribute most information below about 40 km. Particularly, CFC-11 contains considerable information on meridional effective velocities in tropical and midlatitudial regions near 30 km in March–April 2005 (Left upper panel).

Its omission changes these velocities by 20–30%. In contrast, in the region of apparent updraft in northern polar regions, its influence is only about 10%. The sensitivity of horizontal velocities near 30 km to CFC-11 is confirmed by the September–October 2010 test case.

The effect of the omission of CFC-12 is generally much smaller than of CFC-11 (Fig. 8 middle panels). This does not necessarily mean that this species carries less information but that its information is more consistent with that of the other

species. For the analysis of the major warming event in the northern polar region in March–April 2005 (middle left panel), CFC-12 is, in contrast to CFC-11, more relevant for the inference of vertical than horizontal effective velocities. The same is true for HCFC-22 (lower right panel). In this particular test case, HCFC-22 is particularly important at altitudes from 6–12 km on southern midlatitudes.

$CCl_4$ and $SF_6$ broadly contribute in the same regions as the species discussed before, but their contributions are generally

smaller, because measurement uncertainties are larger for these species and their weight in the inversion is thus lower (Fig. 9, top and middle panels). Except for polar winter conditions, the meridional effective velocities seem to be more sensitive to the omission of these species than the vertical effective velocities. Both in March–April 2005 and September–October 2010, $CCl_4$ contributions are largest to horizontal velocities in the altitude regions of 21–30 km and 6–9 (top panels). The contributions of $SF_6$ are even smaller than that of $CCl_4$. They exceed 10% only in the shearing region at 21 km altitude, 70°N-80° in

March–April 2005 and between 6 and 15 km altitude in southern midlatitudinal and tropical latitudes in September–October 2010.

$H_2O$ provides a considerable amount of information (Fig. 9, lower panels, note the different colour scale due to the large amplitude of values). Its contributions are largest where its gradients are largest, namely in the upper troposphere/lower stratosphere and in the mesosphere. In the subsiding segment of the mesospheric circulation at southern polar latitudes in March–

April 2005 its contribution exceeds 50% in some places. In September–October 2010, when the subsiding segment of the mesospheric circulation is situated at northern polar latitudes, the contribution of $H_2O$ even reaches 100%. $H_2O$ also provides important information at tropical and midlatitudinal latitudes below 25 km.

$N_2O$ contributes considerably in the entire altitude range (Figs. 10, upper panels; note the large range of values represented by the colour scale). In wide parts of the atmosphere its contribution exceeds 50%, particularly in March–April 2005.

$CH_4$ and CO provide the bulk of information on the circulation in the upper stratosphere and mesosphere (middle and lower panels). There, contributions exceed 50% in wide regions. However, similar as $N_2O$, they do also provide a lot of information

at lower altitudes, which can hardly be appreciated due to the large range of values represented by the colour scales of the figures.

Overall, the effects of omission of certain species are generally minor to moderate and confined to specific regions, except for the upper stratosphere and mesosphere, where only a few species carry information, *viz.*, $H_2O$, $N_2O$, $CH_4$, and CO. The robustness of the inversion with respect to the omission of single species up to about 40 km indicates that either the MIPAS mixing ratio fields are not biased or that ANCISTRUS is not overly sensitive to such biases. Since a major amount of information exploited by ANCISTRUS is not contained in the mixing ratios themselves but the mixing ratio differences, biases, if existing, tend to cancel out.

One might argue that inclusion of species which contribute only little information, such as $SF_6$ or $CCl_4$, is useless. Admittedly the information provided by these species does not change the results very much. However, inclusion of these species reduces the estimated uncertainty of the retrieved effective velocities. Figure 11 shows the estimated standard deviations, representing the uncertainty of the retrieved horizontal (left panels) and vertical (right panels) velocities due to the propagated uncertainties of the mixing ratio fields, for an ANCISTRUS run with all gases included (top panels), and without $CCl_4$ (middle panels)[1] The estimated uncertainties are reduced by an appreciable amount, mainly in the lower tropical stratosphere. This is more pronounced for the horizontal than for the vertical velocities. In the tropical middle stratosphere at around 30 km altitude, inclusion of $CCl_4$ increases considerably the altitude region where the standard deviation of $v_\phi$ is below 0.06 degrees per month. For tropical middle stratospheric vertical velocities the altitude range where standard deviations are below 20 m/month increases similarly.

The omission of $N_2O$, chosen as an example of a gas which contributes more information, has a larger impact (Fig. 11, lower panels). Particularly at altitudes between about 30 and 50 km, both at polar and tropical latitudes, the standard deviations are up to a factor of two higher when $N_2O$ is omitted.

## 6   Conclusions

ANCISTRUS is a method to infer stratospheric circulation from measured tracer mixing ratios via the inversion of the 2D continuity equation. The primary area of application of this method is the investigation into the structure and possible changes of the Brewer Dobson circulation. In order to validate ANCISTRUS, a series of tests have been performed. By comparison of its application to steady-state conditions to application with deactivated chemical sinks, the contributions of two information pathways were isolated. In the steady-state, ANCISTRUS recovers a field of effective velocities which just compensates the chemical sinks by advection. In contrast, the application with the sinks turned off exploits exclusively the information which is contained in the displacement of patterns of mixing ratios. It was shown that both mechanisms are important to retrieve the full picture and that the latter information pathway is particularly important.

Model recovery tests were performed to test if ANCISTRUS is able to retrieve a known assumed field of effective velocities that was used to generate simulated mixing ratio measurements. Up to about 30 km altitude, ANCISTRUS results have

---

[1]A similar test, but with an older ANCISTRUS version, has already been performed by Eckert (2018)

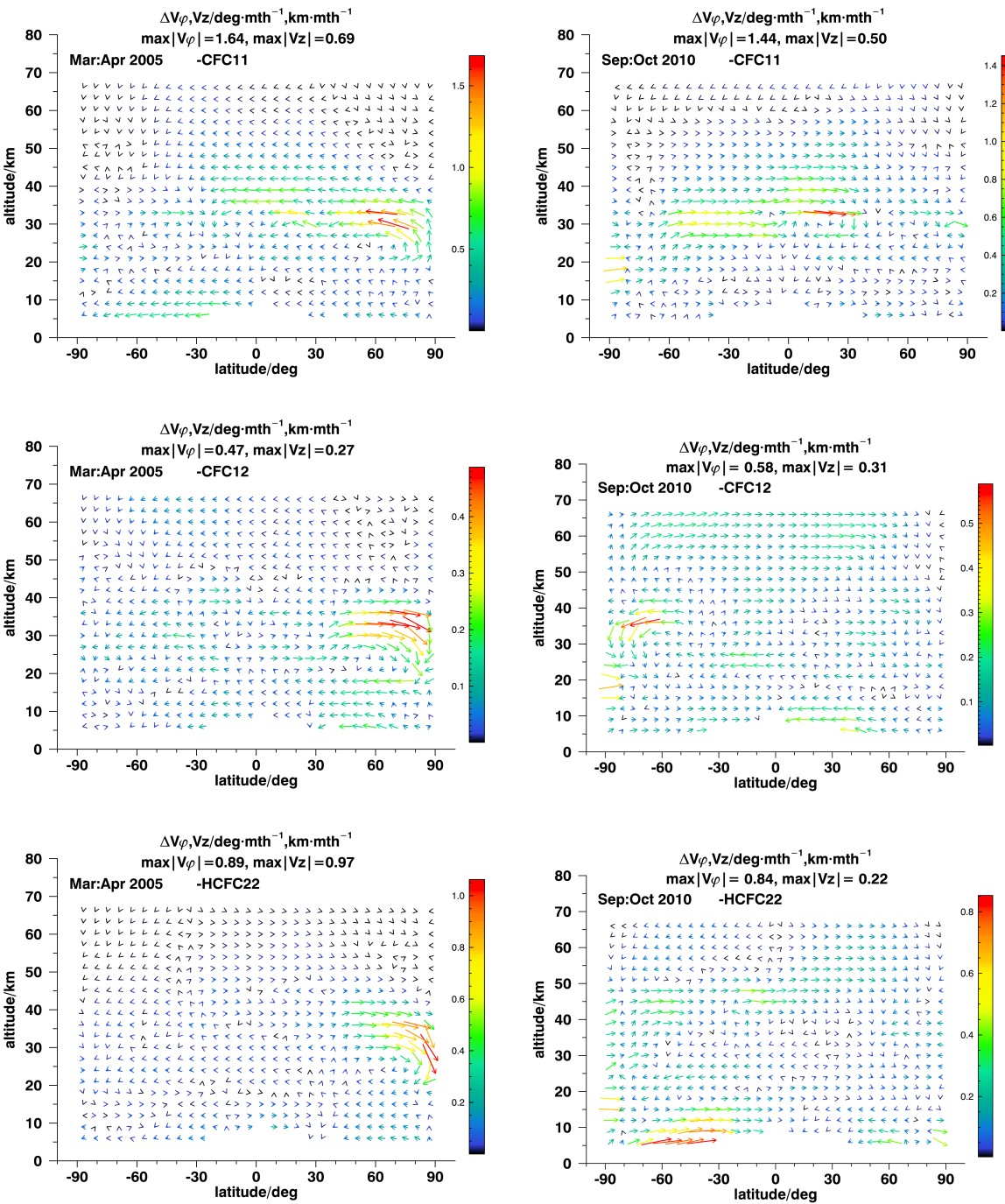

**Figure 8.** : Differences between ANCISTRUS runs with one species omitted and all nine species included for March–April 2005 (left panels) and September–October 2010 (right panels). The missing species are CFC-11 (top panels), CFC-12 (middle panels), and HCFC-22 (bottom panels).

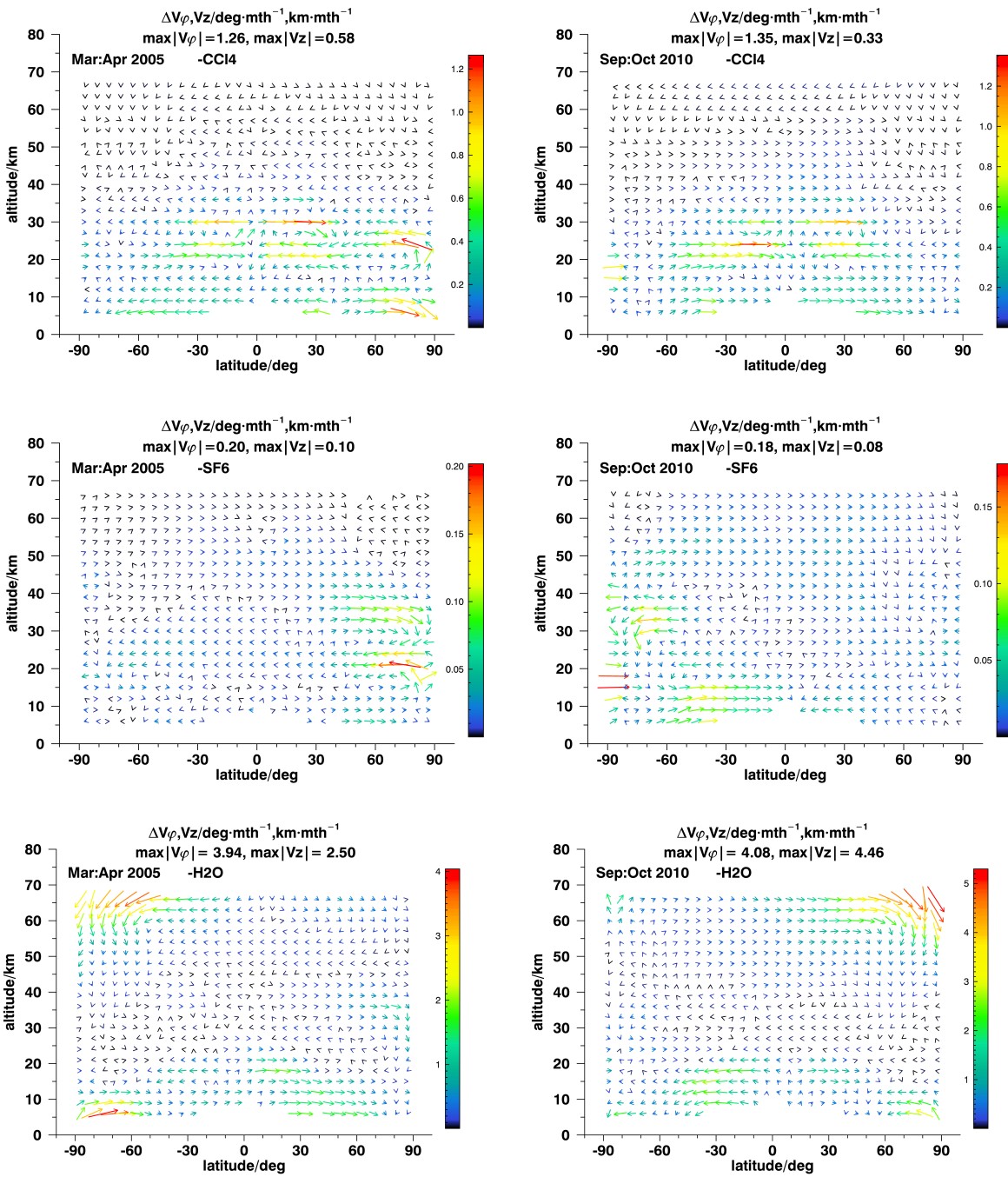

**Figure 9.** : Differences between ANCISTRUS runs with one species omitted and all nine species included for March–April 2005 (left panels) and September–October 2010 (right panels). The missing species are $CCl_4$ (top panels), $SF_6$ (middle panels) and $H_2O$ (bottom panels).

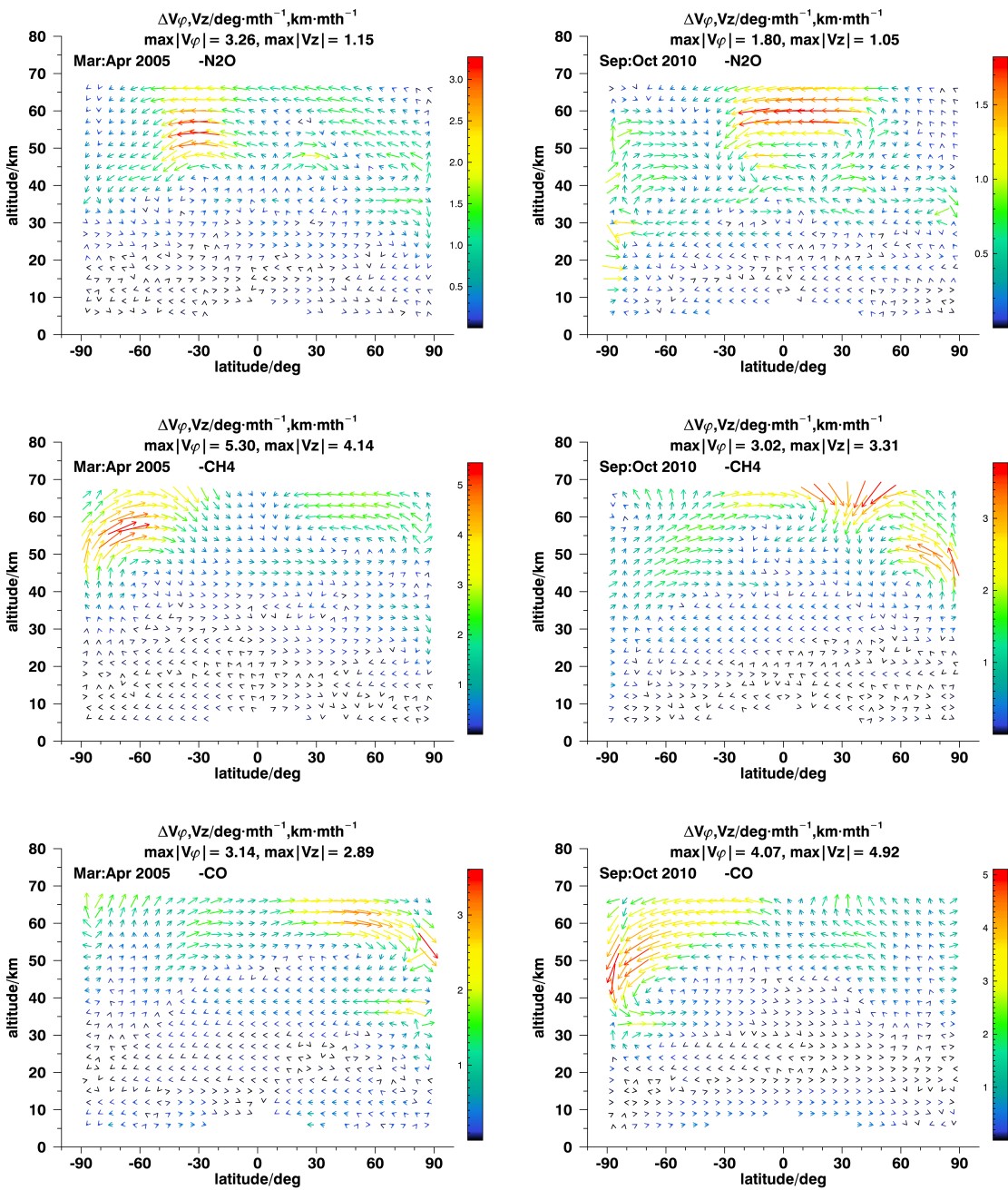

**Figure 10.** : Differences between ANCISTRUS runs with one species omitted and all nine species included for March–April 2005 (left panels) and September–October 2010 (right panels). The missing species are $N_2O$, $CH_4$, and CO.

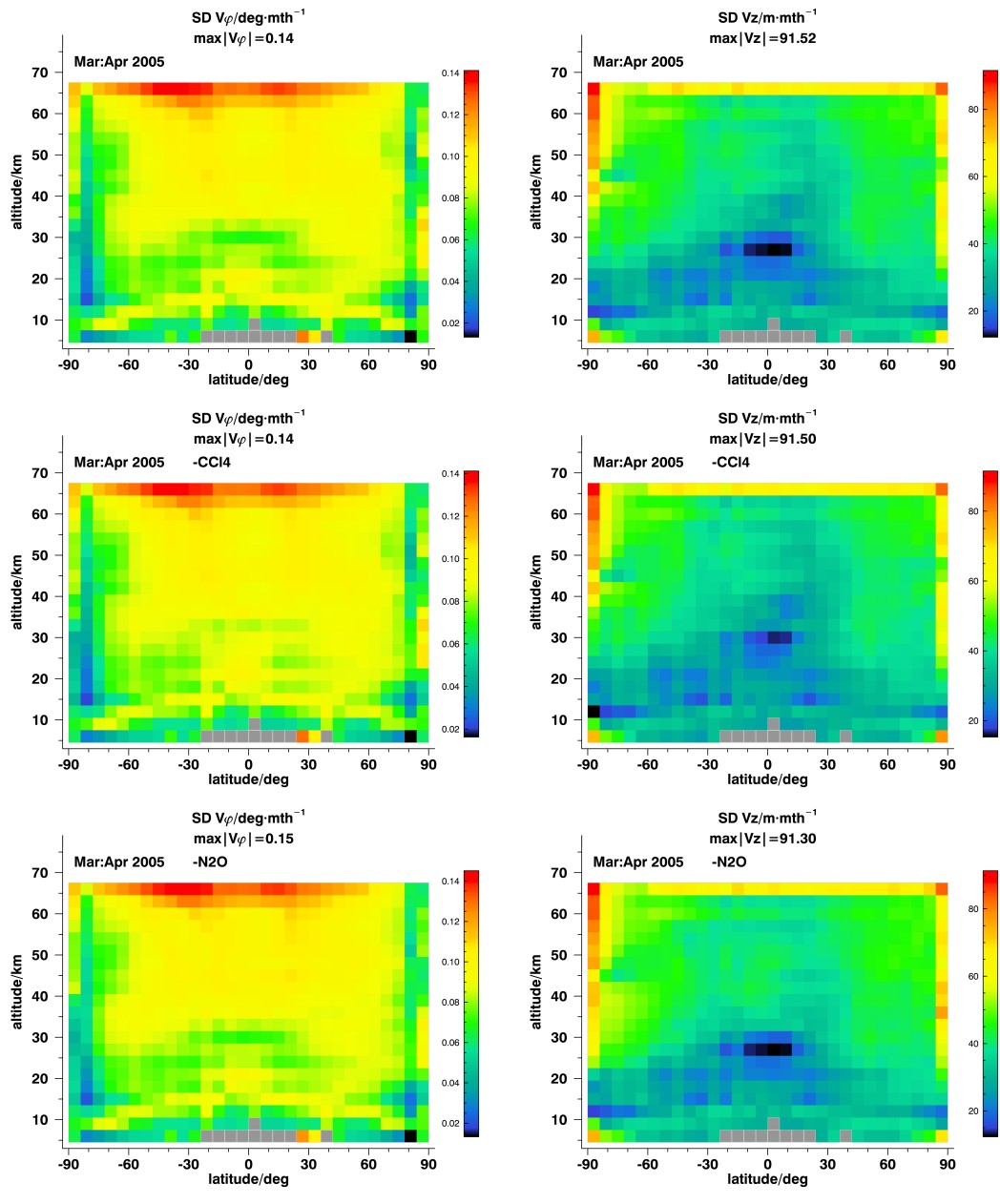

**Figure 11.** : Estimated standard deviations of horizontal (left panels) and vertical (right panels) effective velocities for ANCISTRUS runs with all nine gases (top panel), with CCl$_4$ omitted (middle panels) and with N$_2$O omitted (lower panels) for March–April 2005.

shown to be fairly accurate in a fully quantitative manner. Above, less measurement information is available, and the peak effective velocities deviate from the reference velocities by up to several ten percent. Still structure and patterns are perfectly reproduced and can be considered as robust. Only patterns of very small scales are not resolved. In no case did ANCISTRUS generate artificial structures not present in the reference data. The prevailing underestimation of peak velocities is attributed to the regularization term in the retrieval equation, which pulls values towards zero in the case of insufficient measurement information. The choice of the regularization strength in the ANCISTRUS version tested here was conservative. A rather strong regularization was chosen to avoid ANCISTRUS to produce artificial circulation patterns and to safely achieve convergence of the iteration. According to the terminology of test theory, it had been decided to rather accept type I errors, i.e., to reject a true result, and to safely exclude type II errors, i.e., non-rejection of a false result. The results of this study, however, indicate that there may still be room to fine-tune the regularization in order to achieve less damping of the peak velocities at higher altitudes in a fully quantitative sense. This, however, is deferred to a future paper.

Finally, the information content of the various trace gases used so far in ANCISTRUS applications was investigated. It was found that gases whose omission changes the results only marginally still provide information in the sense that their inclusion reduces the estimated uncertainty of the resulting velocity field. Further, ANCISTRUS proved quite robust with respect to the omission of any single gas. In summary, with respect to the scientific analysis of patterns and structures, we consider the ANCISTRUS algorithm in its current setup as fit for purpose.

*Acknowledgements.* The authors thank Gabriele Stiller for useful comments.

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
