# Peer review of "Direct inversion of circulation from tracer measurements – Part 2: Sensitivity studies and model recovery tests"

_Atmospheric Chemistry and Physics, 2020_

## Referee Comment (RC1) · Anonymous Referee #1 · 29 May 2020

**Direct inversion of circulation from tracer measurements – Part 2: Sensitivity studies and model recovery tests, by Thomas von Clarmann and Udo Grabowski**

**General comments:**

This study presents a new methodology (ANCISTRUS) that provides quantitative information on stratospheric circulation in the form of effective 2D velocity fields obtained from measurements of a set of long-lived trace gases.

The paper presents clear examples of the computed fields and the transport structures they represent. It also provides a valuable illustration of the relative weights that chemical sinks and advection have on the distribution of tracers in the stratosphere. The manuscript also looks into the reliability and sensitivity of the method and shows what regions are better covered by the different chemical species.

The manuscript demonstrates the high potential the new method has to derive effective transport information for the stratospheric region that can complement other existing methods. This will help to overcome information gaps and biases that appear when applying more widely-used existing approaches.

Therefore, this study is a valuable contribution to both the modelling and the observations communities.

In its current form there are, however, several points that need further clarification or development, and editing is also required as detailed here below. Once these aspects have been satisfactorily addressed by the authors I recommend publication of the edited manuscript.

**Specific comments:**

*Some relevant scientific questions should be addressed by the manuscript:*

-To what extent is the ANCISTRUS-derived dataset limited by the biases in the original MIPAS measurements? How are those biases affecting the fields you derive?

-I miss a discussion on the generalisation of the methodology: You have applied the ANCISTRUS method to MIPAS data, but how feasible would it be to apply to other satellite products of atmospheric tracers measurements? Is there any current work on this? How would it be done? This is an important discussion to show the value of the methodology.

-Is the study of the BDC (line 31) the main application of your method or the main application in this study? Are there other applications?

-Sentence 39-41, if I understand correctly, could indicate a limitation of the methodology rather than stability: can it be that for every year the obtained circulation patterns are the same? What about interannual variability, how does your method account for this?

-More information should be included on how the method copes with, and provides information on, the diffusive and dispersive characteristics of transport.

-Why have you chosen years 2005 and 2010 for this study? The manuscript should justify this choice over other months/years in the period covered by MIPAS observations. How representative are these 2005 and 2010 months for the rest of the period?

-How much does the lack of longitudinal information affect the degree of realistic variability in your results?

- If your method does not consider SF6 sinks (line 64) how can you overcome the biases caused by the mesospheric sink of SF6 when using mean age-of-air methods? This needs to be clearly explained and justified as the manuscript claims this is one of the main advantages of the ANCISTRUS method in the study of the BDC.

-Line 105: It is not clear what you mean by "true velocity fields are not known". Even if true velocity fields are not measured, can you use operational analyses from NWP models as a close to reality alternative? What are the assumed velocity fields you used in the tests discussed in this paragraph? How would the use of different assumed velocity fields affect your results and the spurious data you obtained at the boundaries?

-Lines 120-130: Why a climate model? Why not a CTM driven by operational NWP analyses, then one has the tracers distributions obtained from the CTM and the operational velocity fields used to force the simulations?

-Line 152: Is there any alternative regularization that can better resolve adjacent opposite circulation branches? Have you tested it?

*Some parts of the manuscript need substantial editing:*

-The Introduction Section needs to be rewritten: The scope and context for this research is not clearly introduced. Context should be added to the Introduction. Have other similar applications of inverse modelling been previously attempted? What are the reasons to develop this new approach? What are the advantages of the current one compared to previous ones?

Also, some important aspects covered by the paper are not included in the Introduction, e.g. results shown in Section 2, where the effects of sources/sinks and advection are compared. These are very relevant but the Introduction does not say anything about this being an objective of the paper, this information should be added to Section 1.

-Model recovery tests: what this means needs to be clearly explained early in the text.

-Results shown in Figures are very interesting but on several occasions they are not sufficiently explained/developed in the main text. An example is Fig 14: lines 217-220 should give more quantitative information on the amount of uncertainty the inclusion of CCl4 contributes to reduce, as well as explain why it does so more for the vertical field than for the horizontal one.

-The Conclusions Section needs careful revision. This section should be understandable on its own as a Section that summarises the paper. Adding initial sentences summarising ANCISTRUS and why it was developed would improve its readability and completeness. Overall, statements in this section are not clearly backed up, a clearer reference to the results you have shown should be included.

**Technical corrections:**

Line 4: Model recovery tests – a brief explanation/definition would be helpful here.

Lines 8-9: These two sentences should be merged to make the meaning clearer.

line 17: citation here shouldn't use parenthesis

Line 22: "Similar as in other applications of inverse modelling…" some citations to reference previous work and add context to this paragraph should be mentioned.

Line 25: Please consider including some brief information on sinks here, for completeness of text.

Line 37: Change "An application of" to "Applying"

Line 39: Please develop "and so forth" to better understand what you mean here.

Paragraph 41-43 cannot be clearly understood in its present form. Please rewrite.

Line 45: "…confidence in…"

line 51: Sentence needs rewriting. The word intuitively is confusing, the mechanisms described are those providing information to ANCISTRUS, it is not an intuition.

Line 64: "due to its long stratospheric lifetime, SF6 is considered as inert in the given analysis range." If your method does not consider SF6 sinks how can you overcome the biases caused by the mesospheric sink of SF6 when using mean age-of-air methods? This needs to be clearly explained and justified as the manuscript claims this is one of the main advantages of the ANCISTRUS method in the study of the BDC. (See my Specific comment on this).

65-68: This paragraph cannot be clearly understood in its present form. Please rewrite and clarify.

Line 70: "at a certain point at one day". Change point to location; delete 'at'

Line 73: Do you mean "in the real atmosphere"?

Figure 1 caption: for clarity, spell out month in the units (deg month-1)

line 81: "not so much interested in the explanation of the atmospheric features", but this is the main scope of the methodology, right? If you do this in the companion Part 1 paper at least you should mention that here.

Line 85: delete "broadly speaking" or substitute by "on first approximation"

Line 88: (right panels) does not correspond to figures layout

Line 90: "regardless if sinks are estimated.." to "regardless of sinks being estimated.."

Line 99: More information should be included on how the method copes with, and provides information on, the diffusive and dispersive characteristics of transport. (also Specific Comment)

Line 101 and 120: The word severe is not the best one here, perhaps exhaustive, strict, tough..?

Most of page 6, if I understand correctly, is mainly a summary of results in vCG16. If vCG16 shown the validity of the method, this should not be repeated here in a lengthy way, but perhaps written in a way that is more clearly related to the results you show in the current study, e.g. linked to the arguments you use to choose further tests.

Line 140: "September–October 2005" does not correspond to what Fig 3 labels indicate. Please resolve.

Figure 3 and related discussion: what you mean by reference fields needs to be more clearly explained in the main text.

Lines 143-144: how do these underestimation values compare to biases/uncertainties obtained with other methodologies?

Line 149: Please check labels of Figures and corresponding references in the main text match each other.

Line 151: "..are underestimated by about 25% but broadly speaking, the inversion is successful also in quantitative terms." Not clear what you mean, a 25% underestimation does not sound like a quantitative success. Please rephrase or explain further.

Line 155: Move "(Figs 4 and 6)" somewhere else within the sentence, it is not clear whether these two figures refer to the August-September-October 2010 cases or the previous tests.

Line 156: Include some quantitative information on the slight underestimation to put it into context with the results presented earlier. Overall, in the discussion of Figs 3 to 6, more information/explanation should also be included on the reasons for the under/ overestimation of fields.

Line 157: But has it removed existing fields in any occasion? Please add some sentence on this.

Figures 3 to 6 use different color scales for the differences (lower panels in the figs.), wouldn't it be better to use the same color scale to facilitate comparison?

Line 172: Why this particular year?

From results in Figure 7 it seems as if weaker regularization produced better results (middle right panel), why haven't you chosen that regularization instead of the nominal one? If it is due to convergence problems, wouldn't it be useful to show also results for other month/year where the stronger regularization does not work so well?

Line 196-197: If you mean that low sensitivity to the omission of a single species shows the robustness of the methodology, I agree and suggest rephrasing this sentence to make it clearer.

198: "respective" to "corresponding"

Line 199: "similar to a jackknife method", not sure what this means in this context and not sure this part of the sentence is necessary, the set-up is clear.

Line 202: gradients between regions

Some of the Figures 8-13 could/should be combined as multi-panel figures (6 or 9 panels/fig) to reduce the number of Figures and facilitate looking at results in a more straightforward way.

When describing the figures in the main text, some quantitative data should be added, e.g. percentage contribution for each species.

Lines 214-216: If the information coming from the mentioned species contributes to reduce uncertainty, then it is neither useless nor redundant; please consider rewriting these sentences to avoid confusion.

This is also a general suggestion for the whole of Section 5, results in this section show the importance of the different species and the different role they play in forming the final resulting fields, therefore I would suggest not using the word "redundant". Otherwise, why would you use, and show here, redundant information? As far as I understand you have included all species to obtain the final ANCISTRUS results, right? If not, this should be more clearly stated early in the manuscript.

Figure 14: How does the standard deviation responds to the omission of some of the other "minor" species? It would be worth adding one sentence to the main text and perhaps some additional panels to this figure.

Line 221: Please introduce ANCISTRUS at the start of this Section. See also my Specific Comment about Conclusions. Some sentences read as contradictory. For example you say "fairly accurate", then "perfectly reproduced", and then again that there is still room for fine-tuning for a better retrieval of velocities. Overall statements in this section are not clearly backed up, a clearer reference to the results you have shown should be included. The meaning of the last sentence is not fully clear.

---

## Referee Comment (RC2) · Anonymous Referee #2 · 18 Jun 2020

Review of "Direct inversion of circulation from tracer measurements – Part 2: Sensitivity studies and model recovery tests" by T. von Clarmann and U. Grabowski

This manuscript is meant to demonstrate the robustness of the "Analysis of the Circulation of the Stratosphere Using Spectroscopic Measurements" (ANCISTRUS) algorithm described in Part 1 several years ago (von Clarmann and Grabowski, ACP, 2016; vCB16). ANCISTRUS is a continuity equation inversion methodology that relies on monthly differences in trace gas distributions to derive "effective velocities" that describe trace gas transport. I very much appreciate the concept and it would be a great boon to the community if it were demonstrated to be successful in providing information

about the stratospheric transport circulation. The paper is mostly well-written and easy to follow and the model recovery tests and sensitivity tests do indeed demonstrate that the model is relatively robust in terms of being able to reproduce its own results. However, the lead author has referred to this manuscript as a "validation" of the method in the interactive discussion of a second paper under review at ACP (von Clarmann et al., ACP, 2019; vC19), and I find that it falls far short of that description. The model recovery tests, in particular, demonstrate only that the model will retrieve more or less the same effective velocities from more or less the same tracer distribution but do not provide any assessment of whether those effective velocities have any physical meaning or are a unique solution to the continuity equation (these comments are explained in more detail below). If the ANCISTRUS method is to be used to study stratospheric transport in a meaningful way (and the authors indeed attempt use the method to provide a climatology of the meridional circulation in vC19), then those properties must be demonstrated. I therefore cannot recommend publication of this manuscript in ACP without major revisions that address these concerns.

Major technical comments:

Lines 32-33: It has been demonstrated several times (Neu and Plumb, 1999; Linz et al., 2016; Linz et al., 2017) that the age of air is not a good measure of the meridional circulation, but that the age difference between upwelling and downwelling regions is, in fact, equivalent to the diabatic circulation. This methodology does not require assumptions about the age spetra. Certainly if ANCISTRUS were able to successfully retrieve the BDC then it would have some advantages over the age difference, but to compare it to the use of age itself as a circulation diagnostic is somewhat disingenuous.

Lines 39-41: The fact that the interannual variability in the ANCISTRUS-derived circulation is small, particularly in the tropics (from having looked at the figure in vC19), is a red flag for me. We know that the QBO's secondary meridional circulation has a large influence on trace gases in the tropics and subtropics, and any tracer-derived circulation should pick up this variability. It is a very clear signal in trace gas anomalies. Interactive comment

Lines 55-59: I feel that the entire concept of the meridional circulation in this manuscript is highly over-simplified, and this is one example of such over-simplification. The stationarity condition can, in fact, define any number of circulation fields with different mixes of horizontal and vertical advection. In principal, these components might be separable with the right set of trace gases, but there is no evidence presented here that the suite of trace gases used is sufficiently orthogonal to separate horizontal and vertical advection unequivocally.

Lines 69-72: This is another example of over-simplification. The change in amplitude of the structures is also affected by mixing in the real atmosphere. More importantly, while the \*simplest\* (not necessarily best) explanation might be a southward velocity, another explanation would be a shift in the upwelling region (which brings high mixing ratios upward from the tropopause) by 5 degrees south. This would indeed appear as a change in effective southward velocity based on the tracer inversion, but that southward velocity is not a meaningful description of the meridional circulation. In fact, if anything, the effective velocities seem to represent \*anomalies\* in the meridional circulation rather than the circulation itself. The effective velocities are derived from the change in trace gas distributions from one month to the next, but that distribution for any given month already reflects the mean meridional circulation when using real stratospheric trace gases. The familiar shape of tracer isopleths, with an upward bulge in the tropics, strong gradients in the subtropics, relatively flat isopleths in midlatitudes, strong gradients at the vortex edge, and a downward bulge in the vortex are all a reflection of the balance between sinks and the mean meridional circulation and effects of mixing. When you look at the change in this trace gas distribution from one month to the next, it reflects at best the month-to-month change in the circulation, but not the overall circulation itself. All of this highlights the absolute need to understand how well the ANCISTRUS method retrieves an actual circulation field rather than an idealized one (as in Part 1 of the manuscript) or one that it has already generated itself (as in the model recovery tests in this manuscript).
Lines 76-77: I am not sure I agree with the statement that the circulation fields roughly match our expectations of the meridional circulation. For one thing, it is extremely difficult to tell whether this is the case or not from the vector plots. The streamfunction should be plotted instead, with the vectors superimposed over the streamfunction contours if desired. From the plots in the manuscript, the only thing that fairly clearly matches expectations is the circulation are odd (but might more accurately be called interannual differences since two different years are used). I certainly do not clearly see the "branches of the BDC" (line 80, and this phrase should be referenced and defined) – in fact it is hard to see any coherent tropical upwelling region at all. Again, plotting the streamfunction would make the circulation characteristics much clearer.

Lines 77-79: This is another example of an important difference between the effective circulation based on tracers and the BDC. This upward velocity is not meaningful as part of the meridional circulation, which is still downward but weaker than prior to the vortex displacement. Again, it may be more appropriate to view the effective velocity not as a proxy for the BDC, but as anomalies on the background BDC circulation. But this must be demonstrated using an actual circulation field.

Lines 87-94: The plots using annual mean tracer values are, in fact, the only ones that look like the prototypical middle atmospheric circulation to me. The authors seem to indicate that the lack of a pole-to-pole circulation is a deficiency, when, in fact there is no coherent pole-to-pole circulation in the annual mean (nor is there one during the equinoxes, from which the sink terms were used). I also see evidence of the "tropical pipe" ending at ~25 km, where there is strong poleward advection, rather than "reaching up to the mesosphere". The "pipe" is not defined by upwelling, but rather by a lack of communication with the midlatitudes.

Lines 105-112: The authors assert that many tests of this nature were performed for vCG16, but the only ones described or shown used very simplified velocity and tracer fields. What is required is a model recovery test using a realistic meridional circulation
(with vertical and horizontal components and satisfying the continuity equation) and realistic trace gas distributions with both vertical and horizontal gradients. I am not convinced that ANCISTRUS can successfully retrieve a unique solution to the continuity equation that does not alias horizontal and vertical components of the circulation into one another.

Lines 120-130: I am unable to understand why one cannot take the 2-D Transformed Eulerian Mean circulation from a CCM and use it to advect an initial MIPAS trace gas distribution and then retrieve the circulation using ANCISTRUS to see if you recover anything like the model circulation. This would be similar to the tests in vCG16, but using realistic velocity and tracer fields. Some sort of test of this type must be performed before ANCISTRUS can be found to inform our knowledge of the middle atmosphere meridional circulation.

Lines 131-136: As far as I can see, all this demonstrates is that ANCISTRUS is capable of retrieving the same answer when you invert the same field. The effective velocity fields were generated based on the change in trace gases between two months. There is no reason that applying those velocity fields to the initial trace gas distribution should result in a different change in the trace gases than what was used in the initial retrieval, and so for the same distribution, ANCISTRUS essentially gets the same answer twice. This test does not in any way validate that the effective velocities derived are in any way related to actual transport velocities, nor does it demonstrate that the retrieved circulation is a unique solution to the continuity equation that properly resolves both the vertical and horizontal components of the circulation.

Lines 137-139: Even with the reduced vertical scale plots, it is again very difficult to see and interpret these results from vectors. The streamfunction should be plotted, as well as difference plots between the initial and final streamfunction.

Lines 142-143: Again, I do not easily see the "stratospheric branches of the BDC". Please plot streamfunctions and define what you mean be "branches" (I do understand
what is meant, but many readers may not).

Lines 147-148: I'm not sure I agree that the "the slow circulation patterns in the tropopause region and the lower stratosphere are well recovered". If plotted as percent differences, I think some very large discrepancies would emerge.

Lines 153-154: The right panels of Figure 5 are the only figures in the manuscript that seem to resemble the canonical stratospheric meridional circulation. They show rapid poleward transport by the shallow branch (below 15 km here) and strong tropical upwelling, poleward transport, and high latitude downwelling. No other plot shows a coherent upwelling region like this one does. Of course differences are to be expected given the seasonality of the circulation, but the upwelling branch moves back and forth across the year rather than disappearing.

Lines 157-158: While it is true that the second retrieval did not create significantly different patterns than the first, it has not been established that the patterns retrieved in the first place are not artificial given that the algorithm does not appear to have ever been tested with a realistic circulation pattern and realistic tracer distribution.

Figure 3: There are obviously large differences in the velocities at 60S, 60 km for Feb-Mar. Why don't these show up in the difference plots? There are also other examples where the difference plots do not seem to reflect the visual differences between the top and middle plots.

Lines 201-204: Why does withholding CFC11 give the opposite signal to CFC12 in the Arctic? If the sinks are properly accounted for, the effective transport for these two species should be similar.

Line 208-211: I do not understand what is meant by "compressed colour scale". Again, the streamfunction and percent differences might be more useful for seeing the strato-spheric changes.

Lines 211-212: The water vapor "tape recorder" has been extensively used for deriving

**ACPD**
vertical transport in the tropics, yet water seems to do nothing to inform the tropical upwelling. Can this be explained?

Lines 232-233: If this is meant to refer to circulation patterns and structures, then I have to say I strongly disagree that there is evidence that ANCISTRUS is fit for purpose. It does indeed generate a consistent set of patterns and structures from a given set of trace gas fields, but there is no evidence that these patterns and structures are physically meaningful in any way. Until this is demonstrated using a known, realistic circulation field with the MIPAS tracer measurements, I cannot recommend publication of this manuscript.

Minor comments:

Line 1: The wording "allows to infer" is not grammatically correct (it needs a subject). I suggest "provides an inference of".

Line 2: The phrase "both given by" should be "given by both"

Line 4: Using "have shown" in the past tense makes it sound as if these tests were performed in another paper rather than here.

Abstract in general: The abstract does not provide sufficient context for this work or provide any indication of the meaningfulness of the results.

Lines 66-67: The phrase "does effectively not work" should be "effectively does not work"

Line 84: Should "or equatorward transport" be "of equatorward transport"?

Line 88: The reference should be made to "bottom panels" rather than "right panels".

Line 140: I believe "September - October 2005" should be "March - April 2005"

---

## Author Response (AR1)

**Review #1:**

**Comment: General comments:**

This study presents a new methodology (ANCISTRUS) that provides quantitative information on stratospheric circulation in the form of effective 2D velocity fields obtained from measurements of a set of long-lived trace gases.

The paper presents clear examples of the computed fields and the transport structures they represent. It also provides a valuable illustration of the relative weights that chemical sinks and advection have on the distribution of tracers in the stratosphere. The manuscript also looks into the reliability and sensitivity of the method and shows what regions are better covered by the different chemical species.

The manuscript demonstrates the high potential the new method has to derive effective transport information for the stratospheric region that can complement other existing methods. This will help to overcome information gaps and biases that appear when applying more widely-used existing approaches.

Therefore, this study is a valuable contribution to both the modelling and the observations communities.

In its current form there are, however, several points that need further clarification or development, and editing is also required as detailed here below. Once these aspects have been satisfactorily addressed by the authors I recommend publication of the edited manuscript.

**Reply:** The authors thank the reviewer for their encouraging and thorough review.

**Action:** See those related to the specific comments.

**Comment: Specific comments:**

Some relevant scientific questions should be addressed by the manuscript: To what extent is the ANCISTRUS-derived dataset limited by the biases in the original MIPAS measurements? How are those biases affecting the fields you derive?

**Reply:** This is an interesting question. Our main reply is that at first order the ANCISTRUS scheme is fairly insensitive against such biases. The reason is this. ANCISTRUS exploits mainly the gradient information but only to a limited extent the trace gas absolute concentratios. The calculation of gradients involves differences, and biases cancel out when these differences are calculated. Surviving higher order effects are small. We learn this from the fact that the results do not change in any substantial way when a certain species is discarded from the analysis. If a possible bias had a large effect, the neglect of the related species should substantially change the results, which we do not observe.

**Action:** We have included a comment in the Section on the Sensitivity (jack-knive) tests: "Overall, the effects of omission of certain species are generally minor to moderate and confined to specific regions, except for the upper strato-sphere and mesosphere, where only a few species carry information, *viz* $H_2O$, $N_2O$, $CH_4$ and CO. The robustness of the inversion with respect to the omission of single species up to about 40 km indicates that either the MIPAS mixing ratio fields are not biased or that ANCISTRUS is not overly sensitive to such biases. Since a major amount of information exploited by ANCISTRUS is not contained in the mixing ratios themselves but the mixing ratio differences, biases tend to cancel out."

**Comment: I miss a discussion on the generalisation of the method-ology: You have applied the ANCISTRUS method to MIPAS data, but how feasible would it be to apply to other satellite products of atmospheric tracers measurements? Is there any current work on this? How would it be done? This is an important discussion to show the value of the methodology.**

**Reply:** Indeed we plan to apply this method to other satellite data sets. How-ever, the number of suitable data sets is rather small. Global altitude resolved distributions of a sufficient number of species are needed. Candidate data sets under consideration are those of MLS and of ACE-FTS. The challenge with MLS is that the number of suitable species is quite limited ($H_2O$, CO, methyl chloride and $N_2O$ are the most promising ones). A research proposal on this application is actually under evaluation. ACE-FTS offers data of a large num-ber of suitable species. The challenge is the less than optimal global coverage that can be obtained with an instrument measuring in solar occultation.

**Action:** We have added at the end of the fourth paragraph of the Introduction: "Application to trace gas distributions obtained from other satellite missions, such as the Microwave Limb Sounder (MLS, Waters et al., 2006) or the Atmo-spheric Chemistry Experiment – Fourier Transform Spectrometer (ACE-FTS, Bernath et al., 2005) is under consideration."

**Comment: Is the study of the BDC (line 31) the main application of your method or the main application in this study? Are there other applications?**

**Reply:** Indeed the study of the BDC is the main application of the method. Other applications are thinkable but have not yet been studied.

**Action:** None; everything on other applications would, for the time being, be very speculative. Thus we prefer to be quiet about possible other applications.

**Comment: Sentence 39-41, if I understand correctly, could indicate a limitation of the methodology rather than stability: can it be that for**

**every year the obtained variability, how does your method account for this?**

**Reply:** We do see interannual variability, and we see it exactly where we expect it. But the general patterns (e.g., occurrence and directions of high velocities) are reproduced for each year. This issue is discussed in more depth in von Clarmann et al. (2019). The method does not have to explicitly account for the interannual variability because the inversion for one month in one year is fully independent from the inversion of this month in another year. The similarity is not caused by any artificial constraint, we do not constrain the solution to any expected pattern. The similarity is caused by the measurements used. These reflect the seasonality of the circulation.

**Action:** We have added to the text:"[Furthermore, results proved to be stable in the sense that for each year] – within the expected range of variability – [similar circulation fields were found for any particular time of the year, although the estimates were independent from each other.]"

**Comment: More information should be included on how the method copes with, and provides information on, the diffusive and dispersive characteristics of transport.**

**Reply:** The 2D-velocities we present are effective velocities which include the effects of atmospheric diffusion and eddy mixing. How these effective 2D-velocities are related to 3D-velocities and mixing is discussed in the appendices of von Clarmann and Grabowski (2016) and von Clarmann et al. (2019). Dispersion we understand is not a characteristic of atmospheric transport but an effect of a numerical transport scheme. This has been analyzed in von Clarmann and Grabowski (2016). There we also discuss the use of the MacCormack transport scheme, which is simple enough to be operated in the framework of an inverse scheme but still not too susceptible to numerical diffusion and dispersion. In the Section on model recovery tests, previous tests are shortly summarized, and there we say about the transport scheme: "Diffusive and dispersive characteristics can be tested by analysis of the size of the transported mixing ratio maximum and side wiggles created during the transport ... Neither indication of any malfunction nor otherwise conspicious features were found in a long series of these forward model tests" We think that adding more on this would lead to unnecessary redundancies between the papers.

**Action:** None.

**Comment: Why have you chosen years 2005 and 2010 for this study? The manuscript should justify this choice over other months/years in the period covered by MIPAS observations. How representative are these 2005 and 2010 months for the rest of the period?**

**Reply:** As can be seen in von Clarmann et al. (2019), these years are representative for the respective months in other years. The results for a certain month are structurally very similar over the years but vary in a quantitative sense. ANCISTRUS does see the typical expected seasonal patterns. Sometimes a certain pattern may occur a little sooner or later, and the strength of a pattern may vary from year to year (we actually see, e.g., a QBO effect, which is the topic of a separate study). The choice of the test cases was influenced by technical considerations. For example, to do the Jackknife test, it is necessary to start with a month for which data of all gases were available, which was not always the case. Furthermore the Sep-Oct 2010 has a simpler structure and higher velocities while Mar-Apr 2005 has lots of detail structures. Thus we consider these examples as adequate test cases.

**Action:** We have added: " The choice of these years has no particular reason; the seasonal behaviour of these years is well representative for that of the other years available. The months March–April and September–October were chosen because the velocity fields are more structured than in other times of the year and thus more interesting for test purposes."

**Comment: How much does the lack of longitudinal information affect the degree of realistic variability in your results?**

**Reply:** The BDC is a 2D phenomenon. ANCISTRUS in its current implementation does not provide longitudinal information, but we think that this is adequate for studies of the BDC. The problem is rather that, due to correlations of 3D velocities with concentrations, our effective 2D-velocities must not be conceived as zonally averaged 3D meridional velocities. This issue is discussed thoroughly in von Clarmann and Grabowski (2016) and von Clarmann et al. (2019).

**Action:** We have added at the end of the first paragraph of the introduction: "The relationship of these effective 2D velocities to 3D velocities is discussed in the Appendices of vCG16 and von Clarmann et al. (2019, henceforth vC19)."

**Comment: If your method does not consider $SF_6$ sinks (line 64) how can you overcome the biases caused by the mesospheric sink of SF6 when using mean age-of-air methods? This needs to be clearly explained and justified as the manuscript claims this is one of the main advantages of the ANCISTRUS method in the study of the BDC.**

**Reply:** Here we have two arguments:

1. The sinks of $SF_6$ are most relevant above the altitude range we work with. Age-of-air based methods are sensitive to the accumulated loss of $SF_6$ during the air parcel's journey from the stratospheric entry point via the mesosphere and back to the stratosphere. In contrast, ANCISTRUS,

for species without a sizeable stratospheric sink like $SF_6$, uses only the gradient information at two points within the stratosphere. Stratospheric sinks of $SF_6$ are negligibly small. This holds *a fortiori* for time scales as short as one month. The gradient information we use is calculated from measured concentrations. Thus, if such an air parcel contains air depleted in $SF_6$, this is implicitly considered in the gradient calculation. And having the gradients right, the tendency formulation of the continuity equation for $SF_6$ does not depend on sinks outside the domain to which it is applied.

2. Beyond this, the influence of $SF_6$ on the results is surprisingly small. This is because the measurement errors of $SF_6$ are much larger than those of the other species we use. In the inversion more weight is given to the species where the standard error of the zonal mean is smaller. Thus, an effect of $SF_6$ sinks would not distort our results in any appreciable manner, even if there was one. Contrary to that, age-of-air based methods depend on species without stratospheric sinks and with a close to linear tropospheric trend, and there is not much choice. $SF_6$ is the only species which fulfills this requirement and for which global altitude resolved data are available.

**Action:** We have added: "[Intrusion of mesospheric $SF_6$-depleted air does not cause artificial "overaging" of the air (Stiller et al. 2012, Reddmann et al. 2001, Ray et al, 2017)] because for gases without a stratospheric sink, ANCISTRUS takes all information from mixing ratio differences within the analysis domain and not from the absolute abundances. Age-of-air based methods exploit the measured mixing ratio difference between the stratospheric entry point and the measurement location, and the air might have been depleted in $SF_6$ during its potential detour through the mesosphere. The mesospheric loss of $SF_6$ increases the difference and makes the air appear older than it actually is. In contrast, ANCISTRUS exploits the measured differences in the mixing ratios of $SF_6$ between the endpoint and the starting point of a path element of the trajectory only in the domain considered. If the air parcel has re-entered the analysis domain after a possible detour through the mesosphere, any mesospheric loss has affected both the starting point and the endpoint of the path element and thus does not contribute to the difference. [And finally...]

**Comment: Line 105: It is not clear what you mean by "true velocity fields are not known". Even if true velocity fields are not measured, can you use operational analyses from NWP models as a close to reality alternative? What are the assumed velocity fields you used in the tests discussed in this paragraph? How would the use of different assumed velocity fields affect your results and the spurious data you obtained at the boundaries?**

**Reply:** We use velocity fields obtained with ANCISTRUS. For these it is guaranteed that they satisfy the continuity equation and thus belong to the set of

possible solutions of ANCISTRUS. The spurious data we obtained at the boundaries in von Clarmann and Grabowski (2016) were a result of the *ad hoc* choice of a reference velocity field which did not satisfy the continuity equation. Since the continuity equation is hard-wired in ANCISTRUS, it cannot be expected that these fields are reproduced. Analysis fields cannot be directly used because our 2D effective velocity fields represent not only velocities but also atmospheric mixing as well as eddy mixing effects caused by the correlations of velocities and mixing ratios. Furthermore, realistic velocity fields alone do not help because we need also mixing ratio distributions of the relevant species which are consistent with the velocity fields. Most model runs we are aware of either do not cover the full altitude domain or do not provide the distributions of all trace gases needed.

**Action:** The justification of the test scenario has been expanded. See reply to reviewer #2 for detail.

**Comment: Lines 120-130: Why a climate model? Why not a CTM driven by operational NWP analyses, then one has the tracers distributions obtained from the CTM and the operational velocity fields used to force the simulations?**

**Reply:** Our arguments apply equally to a CTM. We have changed the text accordingly. The problem is not how realistic the fields are, as long as they are compliant with the continuity equation, but the problem that our effective 2D-velocities must not be conceived as zonal mean 3D velocities, because they include atmospheric mixing and eddy mixing effects caused by the correlations of velocities and mixing ratios. While such model comparisons are interesting in their own right, the difficulty mentioned makes them less suited in a validation context. Not all available model runs provide distributions of all gases required, and those which do provide them often disagree with the MIPAS measurements. In this case it is unclear which uncertainty is to be assigned to the mixing ratios. This easily tips the balance between the observation error covariance matrix and the regularization. Thus, any occurring differences cannot easily be interpreted in terms of validation. Nevertheless, we have the application of ANCISTRUS to model fields on our agenda for future work.

**Action:** Added: "[On the face of it, tracer and velocity fields from a] chemistry [climate model] or a chemistry-transport-model [would serve the purpose. The comparison of ANCISTRUS results with those] from such a model, [however, suffers...]"

**Comment: Line 152: Is there any alternative regularization that can better resolve adjacent opposite circulation branches? Have you tested it?**

**Reply:** We use a Tikhonov-type regularization which reduces the differences between adjacent vectors of effective velocity. The advantage of this regularization is that the solution can be conceived as a smoothed but otherwise unbiased representation of the truth. The obvious alternative would be a stochastic regularization in the sense of 'optimal estimation'. This is, however, not neutral because the result is biased towards the assumed a priori information. Thus we stick with the Tikhonov-type regularization. After this decision we still have the choice of the strength of the regularization. Strong regularization will degrade the spatial resolution of the result (and thus lead to smaller absolute velocity values at the peak areas) while weaker regularization can give rise to numerical instabilities showing up as unphysical oscillations and leads more often to non-convergence. The optimal choice is application-dependent. For a single case study weaker regularization may be more appropriate because it reveals more details, while for a multi-annual climatological study (currently the focus of our interest) frequent cases of non-convergence shall be avoided and the slightly stronger regularization thus seems more adequate.

**Action:** Added at the end of the Regularization Section: "The optimal choice of the regularization strength, however, is application-dependent, and for particular case studies where convergence turns out not to be a problem, a weaker regularization may be more adequate."

**Comment: Some parts of the manuscript need substantial editing: The Introduction Section needs to be rewritten: The scope and context for this research is not clearly introduced. Context should be added to the Introduction. Have other similar applications of inverse modelling been previously attempted? What are the reasons to develop this new approach? What are the advantages of the current one compared to previous ones?**

**Reply:** Agreed; we will expand on this. We still try to be concise because these issues are discussed at length in vCG16 and vC19, and are already summarized in the third paragraph of the manuscript.

**Action:** We start the first paragraph of the intro with: "Traditionally, the observational analysis of the strength of the Brewer-Dobson circulation relies on the concept of the mean age of stratospheric air (AoA, Waugh an Hall, 2002). The AoA is the average transport time of an air parcel from the stratospheric entry point to the measurement location and is estimated from the mixing ratio of an age tracer such as $SF_6$. As an alternative method, von Clarmann and Grabowski (2016, henceforth abbreviated vCG16) derives..."

**Comment: Also, some important aspects covered by the paper are not included in the Introduction, e.g. results shown in Section 2, where the effects of sources/sinks and advection are compared. These are very relevant but the Introduction does not say anything about this being an objective of the paper, this information should be added to Section 1.**

**Reply:** When we tried to implement this suggestion we have discovered that the manuscript already read :"Since chemical decomposition has been newly implemented in the most recent ANCISTRUS version, the effect of the consideration of sinks is investigated in Section 2". We have decided to expand on this and add:

**Action:** "[... investigated in Section 2.] The purpose of this investigation is to find out how much information on the circulation is provided by the sinks and how much is provided by the displacement of mixing ratio patterns.

**Comment: Model recovery tests: what this means needs to be clearly explained early in the text.**

**Reply:** Agreed.

**Action:** Added: "[... validate the inverse method by model recovery tests.] For these tests, mixing ratio distributions are modeled using known effective velocities. These mixing ratio distributions are then fed into ANCISTRUS to test how well the initial velocity field is recovered (Section 3)."

**Comment: Results shown in Figures are very interesting but on several occasions they are not sufficiently explained/developed in the main text. An example is Fig 14: lines 217-220 should give more quantitative information on the amount of uncertainty the inclusion of CCl4 contributes to reduce, as well as explain why it does so more for the vertical field than for the horizontal one.**

**Reply:** Agreed; the discussion of the figures, and particularly Fig. 14, has been extended. We do not see that the inclusion of $CCl_4$ provides more information for the vertical field than for the horizontal one. By the way: we have found a technical inconsistency in the model recovery plots. ANCISTRUS can work in different operation modes: in its original version, the length of the time-step is calculated from the nominal data, e.g. from 15 March to 15 April. Later versions allow for correction of sampling irregularities in the measured data and use the average time of the actual measurements. In the original model recovery tests both options were unintentionally mixed, which led to additional minor discrepancies in the results. This has been fixed in the revised version.

**Action:**

**Figs. 1 and 2:** We have slightly reorganized the part where we describe which features in the top panel match our expectations;

**Figs. 3 and 4:** The description of the Feb-Mar 2010 case has been slightly expanded;

**Figs. 5 and 6:** Some additional description has been added.

**Fig. 7** Some description and references to the individual panels of the figure have been added.

**Figs. 8...13 (old):** We have reorganized these figures. We have now chosen a two-column format. Now the right panels show the same gases as the left panels, but for September-October 2010. This serves better the discussion of the contributions of the different species. The discussion of these contributions has been slightly expanded.

**Fig 14 (old); new Fig. 11:** We have added some text: [... mainly in the lower tropical stratosphere.] This is more pronounced for the horizontal than for the vertical velocities. In the tropical middle stratosphere at around 30 km altitude, inclusion of $CCl_4$ increases considerably the altitude region where the standard deviation of $v_\phi$ is below 0.06 degrees per month. For tropical middle stratospheric vertical velocities the altitude range where standard deviations are below 20 m/month increases similarly."

Where necessary, plots from Fig. 4–Fig 7 have been replaced with those from a self-consistent test setup, and respective minor changes have been applied to the text.

**Comment: The Conclusions Section needs careful revision. This section should be understandable on its own as a Section that summarises the paper. Adding initial sentences summarising ANCISTRUS and why it was developed would improve its readability and completeness. Overall, statements in this section are not clearly backed up, a clearer reference to the results you have shown should be included.**

**Reply:** Agreed, we will add some general information to make this section more stand-alone, and we have now linked our conclusions better to the results of the main part of the paper.

**Action:** We have added in the beginning of the section: "ANCISTRUS is a method to infer stratospheric circulation from measured tracer mixing ratios via the inversion of the 2D continuity equation. The primary area of application of this method is the investigation into the structure and possible changes of the Brewer Dobson circulation. In order to validate ANCISTRUS, a series of tests have been performed. By comparison of its application to steady-state conditions to application with deactivated chemical sinks, the contributions of two information pathways were isolated. In the steady-state, ANCISTRUS recovers a field of effective velocities which just compensates the chemical sinks by advection. In contrast, the application with the sinks turned off exploits exclusively the information which is contained in the displacement of patterns of mixing ratios. It was shown that both mechanisms are important to retrieve the full picture and that the latter information pathway is particularly important.

Model recovery tests were performed to test if ANCISTRUS is able to retrieve a known assumed field of effective velocities that was used to generate simulated mixing ratio measurements. [Up to about 30 km altitude, ...(here follows the part on model recovery tests from the original conclusion)"]

After this part, we have added "Finally, the information content of the various trace gases used so far in ANCISTRUS applications was investigated. It was found that gases whose omission changes the results only marginally still provide information in the sense that their inclusion reduces the estimated uncertainty of the resulting velocity field. Further, ANCISTRUS proved quite robust with respect to the omission of any single gas. In summary, [with respect to the scientific analysis of patterns and structures, we consider the ANCISTRUS algorithm in its current setup as fit for purpose.]".

**Comment: Technical corrections:**
**Line 4: Model recovery tests  a brief explanation/definition would be helpful here.**

**Reply:** Agreed; we now define this term when first used.

**Action:** See above.

**Comment: Lines 8-9: These two sentences should be merged to make the meaning clearer.**

**Reply:** Agreed.

**Action:** Merged: "Weaker regularization would in some cases allow a more accurate recovery of the velocity fields but there is a price to pay in that the risk of convergence failure increases."

**Comment: line 17: citation here shouldn't use parenthesis**

**Reply:** Thanks for spotting!

**Action:** Corrected.

**Comment: Line 22: "Similar as in other applications of inverse modelling..." some citations to reference previous work and add context to this paragraph should be mentioned.**

**Reply:** Agreed.

**Action:** [Similar as in other applications of inverse modelling,] such as retrieval of atmospheric state variables from radiance measurements (e.g., Rodgers, 2000) or data assimilation (e.g., Ide et al. 1997), [each iteration of the inversion scheme

in ANCISTRUS consists of two steps:...]

**Comment: Line 25: Please consider including some brief information on sinks here, for completeness of text.**

**Reply:** Agreed.

**Action:** [Sinks of trace gases] due to photolysis, OH-chemistry and $O^1D$ chemistry [are considered as described in von Clarmann et al. (2019)...]

**Comment: Line 37: Change "An application of" to "Applying"**

**Reply:** Agreed. As non-native English speakers we appreciate any suggestions to improve the language. Many thanks! (This applies also to other comments in this review).

**Action:** Done.

**Comment: Line 39: Please develop "and so forth" to better understand what you mean here.**

**Reply:** Agreed; we are now more specific here.

**Action:** "and so forth" deleted. The list has been extended instead: "[...expected features like tropical uplift, polar winter subsidence,] stratospheric poleward transport, mesospheric pole-to-pole circulation, [and elevated stratopauses] (vC19)"

**Comment: Paragraph 41-43 cannot be clearly understood in its present form. Please rewrite.**

**Reply:** Agreed.

**Action:** Rewritten: "The ANCISTRUS version used in this paper includes several upates with respect to the original method by vCG16. In particular, [sinks of trace gases are considered] and mixing coefficients are constrained to zero. The latter implies that resulting velocities are effective velocities that also account for the effect of eddy mixing and physical diffusion. [Further details are reported...]

**Comment: Line 45: "...confidence in..."**

**Reply:** Thanks for spotting!

**Action:** 'in' inserted.

**Comment: line 51: Sentence needs rewriting. The word intuitively is confusing, the mechanisms described are those providing information to ANCISTRUS, it is not an intuition.**

**Reply:** What we want to say: there are two idealized, simplified ways to look at this inverse problem, i.e. two ways to understand where ANCISTRUS takes the information from. Our tests show that none of these is fully adequate because ANCISTRUS exploits both.

**Action:** Rewritten: "Two mechanisms link mixing ratio distributions with the circulation and thus allow to retrieve information on the circulation from measured mixing ratio distributions."

**Comment: Line 64: "due to its long stratospheric lifetime, $SF_6$ is considered as inert in the given analysis range." If your method does not consider $SF_6$ sinks how can you overcome the biases caused by the mesospheric sink of $SF_6$ when using mean age-of-air methods? This needs to be clearly explained and justified as the manuscript claims this is one of the main advantages of the ANCISTRUS method in the study of the BDC. (See my Specific comment on this).**

**Reply:** As explained above, this is because ANCISTRUS relies chiefly on measured gradients. For species without a sizeable stratospheric sink like $SF_6$ it relies fully on the gradients. These gradients might be affected by sink reactions in the analysis domain but not by sink reactions above the analysis domain. The age-of-air method is based on the mixing ratio difference between the stratospheric entry point and the target point. If the pathway of the air-parcel involves the mesosphere, this affects the age estimate. In contrast, ANCISTRUS uses the mixing ratio difference between two adjacent gridboxes, and the changes in these gridboxes from one month to the next, which are not affected by the mesospheric sink. If the air was in the mesosphere before, this does not matter, because the history of the air parcel is irrelevant. This is because for all gridboxes involved actual measurements are used. We have expanded on this in the manuscript.

**Action:** See reply to the specific comment above.

**Comment: 65-68: This paragraph cannot be clearly understood in its present form. Please rewrite and clarify.**

**Reply:** Here the same explanation as in the reply to the comment on line 64 applies. In the rewritten text we now refer to the (now more elaborated) explanation above.

**Action:** Replaced by "For reasons discussed above, ANCISTRUS is sensitive only to decomposition of gases within the diagnosed latitude and altitude range

but not to depletion at higher altitudes. Any depletion of, say, $SF_6$ on its way through the mesosphere before it subsides again into the stratosphere thus does not affect the ANCISTRUS results.

**Comment: Line 70: "at a certain point at one day. Change point to location; delete "at"**

**Reply:** Thanks for the correction.

**Action:** Corrected as suggested

**Comment: Line 73: Do you mean "in the real atmosphere"?**

**Reply:** Not quite. This statement is not about what happens in the atmosphere but what of this is essential for ANCISTRUS to correctly reconstruct the fields of efficient velocity.

**Action:** Reworded:"As opposed to both these simplified views where information pathways are assessed in isolation, both mechanisms contribute to the full picture. ANCISTRUS thus exploits both information pathways."

**Comment: Figure 1 caption: for clarity, spell out month in the units (deg month-1)**

**Reply:** Agreed.

**Action:** Changed as suggested

**Comment: Line 81: "not so much interested in the explanation of the atmospheric features", but this is the main scope of the methodology, right? If you do this in the companion Part 1 paper at least you should mention that here.**

**Reply:** This study is meant as a technical validation study. A first step towards the scientific analysis of ANCISTRUS results is reported in von Clarmann et al. (2019). We will mention this.

**Action:** Reworded: "While the scientific interpretation of these fields of effective velocity is provided elsewhere (e.g., vC19), we are, within the framework of this technical study, not so much interested..."

**Comment: Line 85: delete "broadly speaking" or substitute by "on first approximation"**

**Reply:** Agreed.

**Action:** "broadly speaking" deleted.

**Comment: Line 88: (right panels) does not correspond to figures layout**

**Reply:** Yes, indeed. Thanks for spotting.

**Action:** Corrected.

**Comment: Line 90: "regardless if sinks are estimated.." to "regardless of sinks being estimated.."**

**Reply:** Agreed, thanks.

**Action:** Changed as suggested.

**Comment: Line 99: More information should be included on how the method copes with, and provides information on, the diffusive and dispersive characteristics of transport. (also Specific Comment)**

**Reply:** The method provides effective velocities which include diffusive (physical diffusion, not numerical diffusion!) and eddy mixing effects. We understand dispersion to be a numerical effect of transport modelling but not a characteristic of what really happens in the atmosphere. The transport scheme has been tested with respect to its diffusive and dispersive properties in vCG19. The effects of eddy mixing and (physical) diffusion are intentionally aliased into the effective velocities.

**Action:** The respective sentence in the fourth paragraph of the introduction now reads: "In particular, [...] mixing coefficients are constrained to zero. The latter implies that resulting velocities are effective velocities that also account for the effect of eddy mixing and physical diffusion. "

**Comment: Line 101 and 120: The word severe is not the best one here, perhaps exhaustive, strict, tough..?**

**Reply:** 'Severe tests' is a technical term (c.f., e.g., Mayo, 1996). A test is called severe if the likelihood is large that it will refute a hypothesis if false.

**Action:** None.

**Comment: Most of page 6, if I understand correctly, is mainly a summary of results in vCG16. If vCG16 shown the validity of the method, this should not be repeated here in a lengthy way, but perhaps written in a way that is more clearly related to the results you show in the current study, e.g. linked to the arguments you use to**

**choose further tests.**

**Reply:** The first paragraph of the Section is just a short summary of the tests of the forward model. This seems necessary to us, because we rely on these tests and do not test the forward model (transport scheme) again. The second paragraph is a generic explanation of the logic of a model recovery test. We consider it as indispensible to understand the rest of this section. The third paragraph highlights that velocity fields used for model recovery tests must satisfy the continuity equation to allow sensible tests. This argument is essential to understand why we have set up the tests as we did. Neither the second nor the third paragraph are directly related to vCG16. In the fourth paragraph we discuss which options we have to build a sensible test scenario. Only here we shortly come back to vCG16 and we conclude that related tests therein had problems because the test fields did not satisfy the continuity equation. From the fifth paragraph on we describe the test setup chosen for this paper.

**Action:** At the end of the first paragraph we summarize: "We thus consider the transport scheme used by ANCISTRUS as valid."

**Comment: Line 140: "SeptemberOctober 2005" does not correspond to what Fig 3 labels indicate. Please resolve.**

**Reply:** Thanks for spotting!

**Action:** Corrected to "For the March–April 2005 case"

**Comment: Figure 3 and related discussion: what you mean by reference fields needs to be more clearly explained in the main text.**

**Reply:** The reference case is the field of effective velocities used to calculate the trace gas distributions which then are used as "surrogate truth". In this test setup, ANCISTRUS uses these effective velocities to calculate "surrogate measurements" of trace gases. These trace gas distributions are then used to retrieve the velocity field. The comparison of the resulting field and the reference field contains information about the robustness of the ANCISTRUS method. We agree that the logic of model recovery tests and the involved specific terminology should be explained in the text.

**Action:** The fourth and fifth sentence in the second paragraph of this section, where the logic of the model recovery tests is explained, now read "[Such a test is organized as follows.] The assumed velocity field is taken as a reference field and is applied to a measured initial atmospheric state." This sentence now includes a definition of the term 'reference field'. And in the last sentence of this paragraph we replace for clarity "is compared to the one used to simulate ..." with "is compared to that reference field used to simulate...

**Comment: Lines 143-144: how do these underestimation values compare to biases/uncertainties obtained with other methodologies?**

**Reply:** To the best of our knowledge, there exist no other observational methods which provide a picture of meridional middle atmospheric circulation at a spatial and temporal resolution comparable to ANCISTRUS. The $SF_6$-based age-of-air method has uncertainties in the order of a couple of years due to the neglect of the mesospheric sink. Ray et al. (2016) report a difference between $SF_6$-based and $CO_2$-based age measurements of 14-6=8 years, which indicates a bias of more than 100%. Unfortunately no dense global vertically resolved middle atmospheric $CO_2$ measurements are available. To these the sink problem would not apply. Another problem with the use of $CO_2$ as an age tracer is the annual cycle, which causes ambiguities in the transformation of mixing ratios into ages.

**Action:** In order not to compare apples and oranges (i.e. integrated transport times vs. time-resolved transport times) we have decided not to include such a comparison in the paper. Such a comparison would raise more questions than it solves.

**Comment: Line 149: Please check labels of Figures and corresponding references in the main text match each other.**

**Reply:** We have checked this and have not found any inconsistence with respect to the figure references.

**Action:** We have double-checked the figure references.

**Comment: Line 151: "...are underestimated by about 25% but broadly speaking, the inversion is successful also in quantitative terms." Not clear what you mean, a 25% underestimation does not sound like a quantitative success. Please rephrase or explain further.**

**Reply:** Given the possible large bias of the age-of-air based method and the lack of any other method which provides middle atmospheric meridional circulation at a comparable temporal and spatial resolution we find our results not so bad. Furthermore, the numbers quoted do not describe the typical underestimation but features that stand out as particularly problematic. And beyond this, all the patterns are recovered in a very robust way.

**Action:** We now qualify the statement "[Peak velocities in the mesospheric branches of the circulation are underestimated by about 25% but] in large parts of the analysis domain [the inversion is successful also in quantitative terms.]"

**Comment: Line 155: Move "(Figs 4 and 6)" somewhere else within the sentence, it is not clear whether these two figures refer to the**

**August-September-October 2010 cases or the previous tests.**

**Reply:** Agreed.

**Action:** Moved: "Tests for August–September 2010 and September–October 2010 (Figs 4 and 6) confirm ...

**Comment: Line 156: Include some quantitative information on the slight underestimation to put it into context with the results presented earlier. Overall, in the discussion of Figs 3 to 6, more information/explanation should also be included on the reasons for the under/overestimation of fields.**

**Reply:** Agreed to present numbers. The damping of the amplitudes is quite a natural thing when a regularization is used which constrains the differences of values at adjacent gridpoints.

**Action:** The related text has largely been rewritten. See manuscript with Track Changes for details.

**Comment: Line 157: But has it removed existing fields in any occasion? Please add some sentence on this.**

**Reply:** We have observed only one small-scale pattern which has not been recovered.

**Action:** Added: "The small-scale circulation feature at 20°S, 45 km altitude in February–March 2010 is the only instance of a feature in the reference field which has not been reproduced."

**Comment: Figures 3 to 6 use different color scales for the differences (lower panels in the figs.), wouldn't it be better to use the same color scale to facilitate comparison?**

**Reply:** The values in the difference plots are in some cases much too small to be shown in a common color scale with the velocities. For the related discussion it is essential to clearly resolve the differences.

**Action:** None

**Comment: Line 172: Why this particular year?**

**Reply:** We could have chosen almost any year and any month. The only months less suitable for such tests are those where some gases had to be discarded. There is no further particular reason why we have used 2010; September-October is interesting because of the pronounced structures and large velocity contrasts.

With respect to that, Sep-Oct is a particularly severe test which is supposed to react quite sensitive to the choice of the regularization parameter.

**Action:** Added: "[We use September-October 2010 as a test case,] because the large velocity contrasts are a particular challenge for a Tikhonov-type smoothing regularization.

**Comment: From results in Figure 7 it seems as if weaker regularization produced better results (middle right panel), why haven't you chosen that regularization instead of the nominal one? If it is due to convergence problems, wouldn't it be useful to show also results for other month/year where the stronger regularization does not work so well?**

**Reply:** Indeed we find that our chosen regularization is a fair compromise between accuracy and stability. Since currently long-term studies where data gaps are to be avoided are in the focus of our research interest, we have chosen the stronger regularization. For case studies it may be worthwhile to optimize the regularization strength to the particular case. The regularization is not hardwired but a user-defined input and can easily be adjusted to the actual needs. To the second part of the question: Do you mean "where the **weaker** regularization does not work so well"? In these cases we simply have no results. We do not consider the intermediate results of a non-converged iteration as meaningful result. They are simply a data gap.

**Action:** As stated above, we have added at the end of the Regularization Section: "The optimal choice of the regularization strength, however, is application-dependent, and for particular case studies where convergence turns out not to be a problem, a weaker regularization may be more adequate."

**Comment: Line 196-197: If you mean that low sensitivity to the omission of a single species shows the robustness of the methodology, I agree and suggest rephrasing this sentence to make it clearer.**

**Reply:** This is exactly what we mean, and we agree to state this conclusion more clearly.

**Action:** Reworded as suggested: "A low sensitivity to the omission of a single species shows the robustness of the methodology.".

**Comment: 198: "respective" to "corresponding"**

**Reply:** Agreed, thanks.

**Action:** Changed as suggested.

**Comment: Line 199: "similar to a jackknife method", not sure what this means in this context and not sure this part of the sentence is necessary, the set-up is clear.**

**Reply:** The reviewer of von Clarmann et al. (2019) explicitly demanded 'Jackknife' tests to be performed. Thus we prefer to keep this wording.

**Action:** None.

**Comment: Line 202: gradients between regions**

**Reply:** Agreed, thanks.

**Action:** Corrected.

**Comment: Some of the Figures 8-13 could/should be combined as multi-panel figures (6 or 9 panels/fig) to reduce the number of Figures and facilitate looking at results in a more straightforward way.**

**Reply:** Agreed.

**Action:** These figures have been re-organized and combined.

**Comment: When describing the figures in the main text, some quantitative data should be added, e.g. percentage contribution for each species.**

**Reply:** Agreed to provide some quantitative information. However, percentage contributions do not always work because some species make a positive and others a negative contribution (i.e. push the result into the the opposite direction). If the contributions of two gases largely compensate each other and the final velocity is small, each gas would have a very large percentage value, which would be misleading. Further quantitative examples have to be limited to selected examples. The percentage contribution of a gas is different for each gridpoint and each time, thus the full quantitative information cannot be condensed into a few numbers in the text.

**Action:** The discussion of this issue has been largely rewritten and is now more specific. Where appropriate, quantitative information is provided. See the manuscript with track changes for details.

**Comment: Lines 214-216: If the information coming from the mentioned species contributes to reduce uncertainty, then it is neither useless nor redundant; please consider rewriting these sentences to avoid confusion. This is also a general suggestion for the whole of Section 5, results in this section show the importance of the different**

species and the different role they play in forming the final resulting fields, therefore I would suggest not using the word "redundant". Otherwise, why would you use, and show here, redundant information? As far as I understand you have included all species to obtain the final ANCISTRUS results, right? If not, this should be more clearly stated early in the manuscript.

**Reply:** Indeed we have used information of all species. We agree that 'redundant' is misleading in this context.

**Action:** The word 'redundant' is no longer used in the manuscript.

**Comment: Figure 14: How does the standard deviation responds to the omission of some of the other "minor" species? It would be worth adding one sentence to the main text and perhaps some additional panels to this figure.**

**Reply:** Omission of other species has a larger effect. We have chosen $CCl_4$ because for this species the information it provides is least evident from the Jackknife test. For $CCl_4$ we felt the largest pressure to justify why we consider it at all. We agree to add some information on the other species.

**Action:** We have included figures and discussion on $N_2O$ and its effect on the standard deviations.

**Comment: Line 221: Please introduce ANCISTRUS at the start of this Section. See also my Specific Comment about Conclusions. Some sentences read as contradictory. For example you say "fairly accurate", then "perfectly reproduced", and then again that there is still room for fine-tuning for a better retrieval of velocities. Overall statements in this section are not clearly backed up, a clearer reference to the results you have shown should be included. The meaning of the last sentence is not fully clear.**

**Reply:** We agree to include some general information in the conclusion. We do not see a contradiction in our statements. Accuracy refers to the quantitative results, while perfect reproduction refers to the recovery of structure, which is another category. Further, there is nothing principally wrong with results obtained with a strong regularization. They just represent a smoothed version of the true state. The fine tuning does not make the inversion better in a general sense but more adequate for a particular purpose. We will rewrite the conclusion to make this clearer, and we will better link our conclusions to the main part of the paper.

**Action:** Some general information on ANCISTRUS has been added. Now all tests are referred to in the conclusion. The regularization issue is now better

discussed.

**Review #2:**

**Comment: This manuscript is meant to demonstrate the robustness of the "Analysis of the Circulation of the Stratosphere Using Spectroscopic Measurements (ANCISTRUS) algorithm described in Part 1 several years ago (von Clarmann and Grabowski, ACP,2016; vCB16). ANCISTRUS is a continuity equation inversion methodology that relies on monthly differences in trace gas distributions to derive "effective velocities" that describe trace gas transport. I very much appreciate the concept and it would be a great boon to the community if it were demonstrated to be successful in providing information about the stratospheric transport circulation.**

**Reply:** We do not understand why the form of the counterfactual conditional has been chosen here.

**Comment: The paper is mostly well-written and easy to follow and the model recovery tests and sensitivity tests do indeed demonstrate that the model is relatively robust in terms of being able to reproduce its own results.**

**Reply:** We are afraid that the reviewer has misunderstood the model recovery tests. The test did not merely show that the model reproduces its own results. The tests have shown that the algorithm, applied to tracer distributions related to a **known** field of effective velocities does reproduce these. This is a standard procedure in testing inverse methods.

**Comment: However, the lead author [...]**

**Reply:** Does the reviewer suggest that this paper is not co-written by both authors and that its content is not agreed by both authors? Is there any indication of this? Why this *ad personam* comment?

**Comment: [...] has referred to this manuscript as a "validation" of the method in the interactive discussion of a second paper under review at ACP (von Clarmann etal., ACP, 2019; vC19), and I find that it falls far short of that description. The model recovery tests, in particular, demonstrate only that the model will retrieve more or less the same effective velocities from more or less the same tracer distribution [...]**

**Reply:** We disagree. In one case, we use MIPAS tracer fields; in the other case we use tracer fields generated by the model. That these are similar is simply another positive instantiation of the validity of the method. It proves that the velocity fields chosen are actually a solution of the problem. Otherwise the

tracer fields would be quite different from the measured ones.

**Comment: [...] but do not provide any assessment of whether those effective velocities have any physical meaning or are a unique solution to the continuity equation (these comments are explained in more detail below).**

**Reply:** The physical meaning of the effective velocities is quite clear: The resulting effective velocities are those 2D velocities which best reproduce the changes in zonal mean mixing ratio distributions, under consideration of the continuity equation. The physical meaning is identical to age-of-air differences over distance, except that we derive this quantity at better temporal and spatial resolution. Although we apply a lot of diagnostic tools, we have not found any indication of problems with non-unique solutions. For details, see below, under 'specific comments'.

**Comment: If the ANCISTRUS method is to be used to study stratospheric transport in a meaningful way (and the authors indeed attempt use the method to provide a climatology of the meridional circulation in vC19), then those properties must be demonstrated. I therefore cannot recommend publication of this manuscript in ACP without major revisions that address these concerns.**

**Reply:** We have to respectfully disagree. If we understand the reviewer correctly, they say that this discussion paper should be rejected just to block publication of vC19. We do not think that this is the regular reviewing procedure of ACP. This manuscript should be judged by its own content, independently of vC19.
In the manuscript under discussion we have applied the necessary tests to show that ANCISTRUS does exactly what it is supposed to do. The physical meaning of the velocity fields, as provided be the equations containing the transport variables of a 3D atmosphere, is included in the appendices of vCG16 and vC19. If the reviewer would take the effort to look into these appendices, they would better understand what the physical meaning of the effective velocities is. The fact that these are not the same the reviewer is used to is no reason to dismiss this scientific work.

**Comment: Major technical comments: Lines 32-33: It has been demonstrated several times (Neu and Plumb, 1999; Linz etal., 2016; Linz et al., 2017) that the age of air is not a good measure of the meridional circulation, but that the age difference between upwelling and downwelling regions is, in fact, equivalent to the diabatic circulation.**

**Reply:** We do agree that age differences are a useful measure, but they cannot be measured globally without substantial uncertainties. The only global age

measurements are based on $SF_6$; these measurements, however, are strongly biased due to the the mesospheric sink of $SF_6$. This age bias is different in different regions. Thus the age differences as a measure of the meridional circulation will be biased.

**Comment: This methodology does not require assumptions about the age spetra.**

**Reply:** Ploeger and Birner (2016) have shown that age spectra have a strong interannual and seasonal dependence. We think that substracting mean ages associated with different age spectra will also be affected by the differences of the respective age spectra and thus comes down to comparing apples and oranges.

**Comment: Certainly if ANCISTRUS were able to successfully retrieve the BDC then it would have some advantages over the age difference, but to compare it to the use of age itself as a circulation diagnostic is somewhat disingenuous.**

**Reply:** Above we have put forward arguments why age differences are affected by the mesospheric $SF_6$ sink. In the case of the mesospheric $SF_6$ sink, the age differences between different latitudes are even more affected than, e.g., trends at one latitude, as estimated, e.g. by Stiller et al. (2012) or Haenel et al. (2015). We agree that age differences, based on an ideal age tracer, might be an appropriate diagnostic of the circulation in the model world; in the real world, however, where one depends on observational data, this method is deficient, and we thus see no reason why our criticism shall be "somewhat disingenuous".

**Comment: Lines 39-41: The fact that the interannual variability in the ANCISTRUS-derived circulation is small, particularly in the tropics (from having looked at the figure in vC19), is a red flag for me. We know that the QBO's secondary meridional circulation has a large influence on trace gases in the tropics and subtropics, and any tracer-derived circulation should pick up this variability. It is a very clear signal in trace gas anomalies.**

**Reply:** First, we would have expected here a review of this manuscript and not one of vC19. And secondly, we have **not** said that the "interannual variability in the ANCISTRUS-derived circulation is small", but that "for each year similar circulation fields were found for any particular time of the year." We do see, for example, a clear QBO signal in the inferred fields of effective velocity. This QBO signal is currently under investigation but it belongs neither in a technical validation paper nor in a paper which focuses on the climatology in the sense of multi-annual mean circulation.

**Comment: Lines 55-59: I feel that the entire concept of the merid-**

**ional circulation in this manuscript is highly oversimplified, and this is one example of such oversimplification.**

**Reply:** We are afraid that the reviewer grossly misunderstands the purpose of Section 2. We clearly state that here we do **not** describe the "concept of the meridional circulation" but that we investigate two "candidate mechanisms [that] can explain where ANCISTRUS takes the information from..." (line 51). This is not our view of the circulation but an assessment of the sensitivity of ANCISTRUS to the various information sources.

**Action:** The first lines of this Section have been rewritten; see reply to Reviewer #1.

**Comment: The stationarity condition can, in fact, define any number of circulation fields with different mixes of horizontal and vertical advection. In principal, these components might be separable with the right set of trace gases, but there is no evidence presented here that the suite of trace gases used is sufficiently orthogonal to separate horizontal and vertical advection unequivocally.**

**Reply:** The evidence is in that the system of equations has a solution at all. The fact that the condition number stays within reasonable bounds proves that the system of equations we solve has no problem with collinearity. If ambiguity due to insufficient orthogonality was a problem, the inversion would face a singular matrix and we would not get any solution at all. We use all established diagnostics to detect possible ill-posedness of the inverse problem. We do agree that the solution would be ambiguous if we had data at two places only, but we have many data points and the continuity equation has to be satisfied everywhere. Since we do not have only mixing ratios at two points but full vertically and latitudinally resolved distributions of air density and a series of trace gases, the inverse problem is better constrained than one might think. We have by far more equations than unknowns, and we reduce the effective degree of freedom of the system further with the regularization term. Ill-posed inverse problems going along with ambiguous solutions are terribly sensitive to noise and are instable in the sense that infinitesimal changes in the input entail huge differences in the output. We observe the opposite. If the solutions were indeed ambiguous due to the lack of orthogonality, it would not be possible that ANCISTRUS finds similar structures independently for many years. If the inverse problem really was ill-posed, it would be over-sensitive to variations in the mixing ratios. It would produce very different results when we apply ANCISTRUS to, say, the same month of a different year. We observe the opposite. Further, it would not be explainable that patterns evolve smoothly from one month to the next. Also it would not be possible that discarding a gas has only minor effect on the result. Beyond this, non-orthogonality would lead to a solution-space instead of a point-solution. The mathematical and diagnostic tools we use are well established standard and widely used in many fields of science.

Of course the steady state assumption provides less information than the regular case where structures are transported. But this is exactly the point we want to make in this section. This test case is an investigation of this information pathway in isolation. With this test case we show that the idealized steady state assumption does not provide sufficient information to recover the circulation field in full. Thus we do not understand how the criticized lack of information in this test case can be put forward as an argument against ANCISTRUS which, in its normal application, uses both pathways.

**Comment: Lines 69-72: This is another example of oversimplification.**

**Reply:** The isolation of different information pathways as presented in Section 2 is not a simplification but a scientific study in its own right.

**Comment: The change in amplitude of the structures is also affected by mixing in the real atmosphere.**

**Reply:** We do agree, and it is for this reason why we call the velocities effective velocities. By the way, the age of air differences as a measure of the circulation share the same characteristic. Our effective velocities can be conceived as age of air differences per path element, but far better resolved in space and time. If our approach is an "oversimplification" because mixing is aliased into effective velocities, then any age-of-air based method is an oversimplification as well.

**Comment: More importantly, while the simplest (not necessarily best) explanation might be a southward velocity, another explanation would be a shift in the upwelling region (which brings high mixing ratios upward from the tropopause) by 5 degrees south. This would indeed appear as a change in effective southward velocity based on the tracer inversion, but that southward velocity is not a meaningful description of the meridional circulation.**

**Reply:** First, a maximum is characterized by the fact that all values in its horizontal and vertical neighbourhood are smaller. A displacement of such a maximum cannot be caused by a shift in the upwelling region.
Second, we did not expect that the reviewer (or any reader) would take this simple example in the paper literally. We tacitly assumed that it is clear that the continuity equation is satisfied at any point in the system. We analyze the mixing ratio changes at all points simultaneously, and an unphysical velocity vector which would be the simplest solution for one point in the system would cause increased residuals at the other gridpoints. Since ANCISTRUS minimizes the residuals at all gridpoints simultaneously, it would not accept such a solution but search for the global minimum of residuals.

**Action:** We have inserted:"[...this is best explained by a southward velocity of 5 degrees per month,] assuming that this solution satisfies the continuity equation globally.".

**Comment: In fact, if anything [...],**

**Reply:** Is there any evidence that the effective velocities might not represent "anything"? Or is this just a rhetoric trick to dispraise our paper and our method?

**Comment: [...] the effective velocities seem to represent anomalies in the meridional circulation rather than the circulation itself. The effective velocities are derived from the change in trace gas distributions from one month to the next, but that distribution for any given month already reflects the mean meridional circulation when using real stratospheric trace gases.**

**Reply:** We strongly disagree. At places where the change of mixing ratios is zero, the equations provide the information from the balance of advection and sinks; where we have patterns which are transported and thus go along with local changes in mixing ratios, these provide additional information. The patterns themselves may be considered as anomalies, but how these are transported is controlled by the total (i.e. mean plus anomalies) circulation. The signal is imprinted by any – random or seasonal or whatever – effect. The most prominent such effect is the atmospheric water vapour tape recorder. The imprinted signal is an anomaly in the sense that the water vapour amount at the stratospheric entry point has a pronounced seasonality. But how this signal is transported upwards just reflects the total circulation, not only its anomaly. We have many more species than water vapour only, and due to the natural variability of the atmosphere, there is a huge number of anomalies in the mixing ratio distributions. And these patterns are transported, horizontally and vertically; and similarly as the tape recorder, where the ascent of $H_2O$ anomalies provides information on tropical uplift, the displacement of other anomalies we see provides additional information on the actual circulation. All these "additional quasi-tape recorders" contain signal about the total circulation, not only on circulation anomalies.

**Action:** We have added at the end of the second paragraph in this section:"A widely used method that uses this information pathway is the analysis of the ascent rate in the tropical pipe by means of the water vapour tape recorder (Mote et al., 1996)

**Comment: The familiar shape of tracer isopleths, with an upward bulge in the tropics, strong gradients in the subtropics, relatively flat isopleths in midlatitudes, strong gradients at the vortex edge, and a downward bulge in the vortex are all a reflection of the balance between sinks and the mean meridional circulation and effects of mixing.**

**Reply:** Yes, and within ANCISTRUS the interplay between sinks and advection is an important information source. This information, however, is not exploited for annual mean fields but for actual ones and is complemented by the information contained in the pattern transport. ANCISTRUS provides the total actual circulation field and not the steady state field in isolation.

**Comment: When you look at the change in this trace gas distribution from one month to the next, it reflects at best [...]**

**Reply:** What does the reviewer intend to say with the words "at best", and on which evidence is this based?

**Comment: [...] the month-to-month change in the circulation, but not the overall circulation itself.**

**Reply:** If the changes in the trace gas distribution happen to be zero, then we get the velocities which compensate the sinks and which are associated with the steady state. But on top of the steady state the trace gas distributions in the real world change from instance to instance. This is because of the time-dependence of sinks, the time-dependent lower boundary condition, and a natural variability of circulation. As said above, what we get is the total circulation at a certain time. Who denies the information content of pattern transport on the total circulation commits oneself to also deny that the atmospheric tape recorder bears any information on the circulation. What we see is the total actual circulation, composed of the background circulation and its anomalies.

**Comment: All of this highlights the absolute need to understand how well the ANCISTRUS method retrieves an actual circulation field rather than an idealized one (as in Part 1 of the manuscript) or one that it has already generated itself (as in the model recovery tests in this manuscript).**

**Reply:** Here the reviewer seems not to distinguish between the tests of the forward model and the test of the inversion scheme. We rely on Part 1 of the manuscript only for the tests of the forward model. For this purpose, idealized tests are the most severe ones, because diffusive and dispersive characteristics of the transport model show up clearly, and the results can be verified by analytical calculations. We do **not** refer to Part one of the manuscript for the tests of the inversion scheme. The requirement to use an "actual circulation field" is unfulfillable because the actual circulation field is unknown and unknowable. Using ANCISTRUS-derived fields as reference fields guarantees that the reference field satisfies the continuity equation and thus can be recovered by the scheme. Related mixing ratio fields at the end of the time step are **not** the same as used in the first analysis. Thus, the model recovery test is **not** a repetition of the first inversion where MIPAS trace gas measurements have been used for the VMR fields at the second time-step.

Comment: Lines 76-77: I am not sure I agree with the statement that the circulation fields roughly match our expectations of the meridional circulation. For one thing, it is extremely difficult to tell whether this is the case or not from the vector plots. The stream-function should be plotted instead, with the vectors superimposed over the streamfunction contours if desired. From the plots in the manuscript, the only thing that fairly clearly matches expectations is the circulation in the mesosphere,[...]

**Reply:** To us the vector representation is more instructive. We appreciate that different people have different preferences, thus we will make all the data of this paper available via the KITopen portal. Then everybody can plot the data in their own preferred representation.

Comment: [...] though the seasonal differences in the height of the circulation are odd (but might more accurately be called interannual differences since two different years are used).

**Reply:** Does there exist any observational evidence against this altitude difference?

Comment: I certainly do not clearly see the "branches of the BDC" (line 80, and this phrase should be referenced and defined) – in fact it is hard to see any coherent tropical upwelling region at all. Again, plotting the streamfunction would make the circulation characteristics much clearer.

**Reply:** We see a lot of the expected features in, e.g., the top panel of Figure 1:

1. subsidence in the Antarctic in early Austral winter;

2. a small but coherent upward component above the equator (the tropical upwelling is a very slow process; one cannot expect to see it as clearly as, say, polar winter subsidence);

3. poleward velocities at about 20–30 km and above 35 km altitude in the Southern hemisphere;

4. poleward velocities at about 15–20 km altitude at Northern midlatitudinal and polar latitudes;

5. a signal of a sudden stratospheric warming, retrieved for a time period when a sudden stratospheric warming actually had happened.

But all this discussion is only about a little side remark and has little to do with the test we present. The purpose of this figure is to demonstrate how both the

advection-sink balance and the pattern transport contribute to the full picture.

**Action:** Our list of retrieved expected features is now more specific; see reply to review #1 for details.

**Comment: Lines 77-79: This is another example of an important difference between the effective circulation based on tracers and the BDC. This upward velocity is not meaningful as part of the meridional circulation, which is still downward but weaker than prior to the vortex displacement.**

**Reply:** We agree that the physical velocity vectors of an air parcel in a 3D world point downward. The problem is that in the 2D world in a polar coordinate system the displacement of an initially perfectly symmetric vortex off the pole cannot be represented, and there exist no latitudinal 2D velocities that could generate the observed effect of increasing VMRs of most trace species above the pole. To retrieve a velocity which does not exist in the 2D world is too much to ask from a scheme that is based on the 2D continuity equation. The counter-intuitive result does not hint at a problem with ANCISTRUS but it does hint at a problem with any 2D representation of the 3D world. Given the characteristics of the 2D world, ANCISTRUS retrieves exactly the perfect solution, i.e., the only 2D velocity field which is able to reproduce the observed trace gas observations. As we understand that the BDC is a 2D description of stratospheric circulation, we do not quite agree that this is a "difference between the effective circulation based on tracers and the BDC".

**Action:** We have added: "Due to symmetry around the pole, in a 2D representation there is no horizontal velocity which could reproduce this phenomenon. This result, seeming counter-intuitive at first glance, is not a weakness of the ANCISTRUS method but rather a characteristic of the representation of the 3D atmosphere in 2D in general.

**Comment: Again, it may be more appropriate to view the effective velocity not as a proxy for the BDC, but as anomalies on the background BDC circulation.[...]**

**Reply:** We disagree; we do not see anomalies but we see the total 2D-circulation (background plus anomalies), which must, however, not be conceived as the average of the 3D velocities, due to the eddy terms and effects discussed above.

**Comment: [...] But this must be demonstrated using an actual circulation field.**

**Reply:** Actual circulation fields are not available. The reviewer seems to tacitly assume that models represent the truth. Climate models may or may not describe the average state of the atmosphere but not the actual one. Chemistry

transport models are driven by meteorological analyses but are constrained to observations only up to the middle stratosphere. Thus, there is no reason to claim that model fields are closer to the actual atmosphere than our measurements, particularly in the upper part of our analysis domain.

**Comment: Lines 87-94: The plots using annual mean tracer values are, in fact, the only ones that look like the prototypical middle atmospheric circulation to me. The authors seem to indicate that the lack of a pole-to-pole circulation is a deficiency, when, in fact there is no coherent pole-to-pole circulation in the annual mean (nor is there one during the equinoxes, from which the sink terms were used). I also see evidence of the "tropical pipe" ending at25 km, where there is strong poleward advection, rather than "reaching up to the mesosphere". The "pipe" is not defined by upwelling, but rather by a lack of communication with the midlatitudes.**

**Reply:** Figures 1 and 2 are not meant as a discussion of atmospheric processes. They are meant to show that both information paths, advection-sink-balance and pattern transport, are important to reconstruct the full picture. We have included an additional panel in Figures 1 and 2 where we base the advection sink balance of monthly means instead of annual means. Also in this case, the pole-to-pole circulation is mutilated (new Fig. 1, lower left panel) or even absent (new Fig. 2, lower left panel). Since by now no time-resolved global measurements of the meridional circulation were available, it is no surprise that the annual mean example looks more familiar than our time-resolved analyses. Similarly, prior to the invention of the telescope, when the human eye could not resolve the satellites of Jupiter, the prototypical sky was one without Jupiter's satellites.

**Action:** To avoid quibbling about words, we have replaced "tropical pipe" with "tropical upwelling". Further, we have reworded the remaining part of the sentence as follows: "... and the remaining patterns are two rather symmetric transport cells in each hemisphere, the stronger one around 50 km covering all hemispheric latitudes, and a weaker one around 25 km, located in the subtropics." Further, we have included panels in Figures 1 and 2 where the advection–sink balance is based on monthly mean mixing ratio fields.

**Comment: Lines 105-112: The authors assert that many tests of this nature were performed for vCG16, [...]**

**Reply:** This rephrasing of our text by the reviewer does not at all capture what we say in the lines quoted. The reviewer's paraphrasing sounds as if we wanted to suggest that we have made enough tests in vCG16. But what we actually say is quite the opposite. We critically discuss what can be learned from these tests. It is hard to believe that this distorting paraphrasing is unintentional. We consider this as a rhetoric of which the only purpose is to create some animus

against the authors. The wording "assert" seems to suggest that the authors are lying.

**Comment: [...] but the only ones described or shown used very simplified velocity and tracerfields.**

**Reply:** We have to distinguish two cases: The tests of the forward model and the tests of the inverse model.
A transport forward model is best tested with very simple and extreme cases (large gradients and gradient changes in the fields). This is the only reasonable way to test the diffusive and dispersive characteristics of a transport model. With realistic cases multiple effects are superimposed, and we have no reference to compare with. We thus consider our tests of the forward model as severe and valid. And we clearly state that the tests of the inversion tool made in vCG16 were insufficient because the reference fields to be retrieved did not comply with the continuity equation. Since the continuity equation is a hard constraint, large differences between the results and the reference fields were unavoidable. To remedy this deficiency is the main purpose of this manuscript. Thus we do not understand why exactly this deficiency is criticised here. Here (and elsewhere) our arguments are torn out of context to twist our words.

**Comment: What is required is a model recovery test using a realistic meridional circulation (with vertical and horizontal components and satisfying the continuity equation) and realistic trace gas distributions with both vertical and horizontal gradients. I am not convinced that ANCISTRUS can successfully retrieve a unique solution to the continuity equation that does not alias horizontal and vertical components of the circulation into one another.**

**Reply:** The tests we present are based on realistic trace gas distributions and use reference fields of effective velocity that satisfy the continuity equation. As described above, ambiguities between horizontal and vertical components of the circulation would show up in the in very different solutions for slightly different situations, and a failed model recovery test. None of these indicators of ambiguity were encountered in any of our tests. All these diagnostics are established standard.
We have meanwhile model recovery tests available based on the annual mean states (considered as more realistic by the reviewer; not by us, however). ANCISTRUS recovers the velocity fields even better than in the model recovery tests presented in the manuscript. This is because the test cases chosen for the paper were particularly difficult cases with a lot of structure.

**Comment: Lines 120-130: I am unable to understand why one cannot take the 2-D Transformed Eulerian Mean circulation from a CCM and use it to advect an initial MIPAS trace gas distribution and then retrieve the circulation using ANCISTRUS to see if you recover any-**

**thing like the model circulation. This would be similar to the tests in vCG16, but using realistic velocity and tracer fields. Some sort of test of this type must be performed before ANCISTRUS can be found to inform our knowledge of the middle atmosphere meridional circulation.**

**Reply:** MIPAS mixing ratios cannot be combined with modeled velocity fields because these are typically not consistent with each other. As we have learned from the tests in Section 2, the full information is not contained in the difference fields alone because of the sink-advection-balance. To combine CCM 2D velocity fields with MIPAS might be adequate for $SF_6$ which has no stratospheric sinks. Only for $SF_6$ we have $\hat{\vec{v}} = f(\Delta vmr)$, where $\hat{\vec{v}}$ is the estimated field of effective velocity vectors, and where $\Delta vmr$ is the field of mixing ratio differences between the beginning and the end of the time step. $SF_6$ alone, however, is not sufficient to constrain the inverse problem. For gases with stratospheric sinks we have, due to the compensation of sinks by advection, $\hat{\vec{v}} = f(\Delta vmr, vmr_1)$, where $vmr_1$ is the initial velocity field. That is to say, for other concentrations, other velocities are necessary to balance the sinks. The velocity information is not provided by the mixing ratio differences alone. Model velocities are not identical to the real velocities which made the trace gas contributions as they are. Thus, one cannot expect that ANCISTRUS retrieves the modeled velocities, because there is a 'hidden' velocity term in the absolute values of the mixing ratios. From a validation which will result in differences between the result and the reference velocity field which can be explained by such inconsistencies we do not learn anything about the reliability of ANCISTRUS. We need a test setup which allows to unambiguously attribute each discrepancy to ANCISTRUS.

**Action:** Added: "The use of velocities from a model applied to MIPAS volume mixing ratios to generate mixing ratio fields at the second time step does not solve the problem either. The reason is this. As we have learned from the tests in Section 2, the velocities and the initial mixing ratio distributions cannot be chosen independently. For species with sinks in the stratosphere, not only the mixing ratio differences between the beginning and the end of a time step depend on the velocities, but also the absolute concentrations and their spatial distributions. Inconsistencies between the velocity field and the mixing ratio distributions would thus lead to artefacts in the result of the test. A test where it is not possible to decide if any discrepancy between the reference velocity field and the retrieved velocity field is due to this type of artefact or to a possible malfunction of ANCISTRUS is not useful for validation purposes."

**Comment: Lines 131-136: As far as I can see, all this demonstrates is that ANCISTRUS is capable of retrieving the same answer when you invert the same field.**

**Reply:** We are afraid that the reviewer has grossly misunderstood the logic of the model recovery test. The key point is that we need (a) a field of effective velocities satisfying the continuity equation, and (b) tracer fields which are perfectly consistent with this velocity field. We achieve this by generating the tracer fields with our own model. This guarantees that we can attribute all differences between the result and the reference field to our inversion and that there is no other "excuse".

**Action:** The logic of a model recovery test has been described in more depth in reply to review #1.

**Comment: The effective velocity fields were generated based on the change in trace gases between two months. There is no reason that applying those velocity fields to the initial trace gas distribution should result in a different change in the trace gases than what was used in the initial retrieval, and so for the same distribution, ANCISTRUS essentially gets the same answer twice.**

**Reply:** We disagree. If ANCISTRUS was defective, it would not get the same answer twice. If, e.g., ANCISTRUS would alias vertical into meridional velocities, this effect would also be visible when the fields resulting from the tests are compared to the reference fields.
Our test is by no means redundant with the initial inversion. In the initial inversion the mixing ratio distributions were measured ones. In the model recovery test the mixing ratio distributions are calculated ones. Since the system of equations is over-determined, these two cannot be the same. The measurement space is of a far larger dimension than the retrieval space, and in the least squares inversion this excess information is lost, we will not get it back with the forward calculation. The forward model will thus not be able to exactly reproduce the initial, measured, mixing ratio distributions. The fact that the distributions are similar is just another proof that what we got first is indeed a valid solution of the inverse problem.
If something went wrong with the inversion, we would **NOT** get anything similar to the reference velocity distribution in the model recovery test. We do not claim that the model recovery tests are meant as a test of the forward model involved. This has been tested independently in vCG16.

**Comment: This test does not in any way validate that the effective velocities derived are in anyway related to actual transport velocities, [...]**

**Reply:** This is not the purpose of the model recovery test. The testing has been split up into two logical steps. The forward model test in vCG16 provided evidence that the forward model involved models the transport in a realistic manner. The model recovery test provides evidence that a solution consistent with the forward model in use is found, and only both these tests together validate that the effective velocities derived are in anyway related to actual transport velocities. The model recovery test does show that we get the reference velocity field back if we feed ANCISTRUS with the associated mixing ratio data. This is exactly the purpose of a model recovery test, and ANCISTRUS has passed this test. If ANCISTRUS aliased vertical and horizontal velocities when applied to MIPAS data, there is no reason why it should not alias these again when fed with simulated data and cause differences from the reference field of effective velocity.

**Comment: [...] nor does it demonstrate that the retrieved circulation is a unique solution to the continuity equation that properly resolves both the vertical and horizontal components of the circulation.**

**Reply:** We apply all established diagnostics to detect possible ill-posedness of the inverse problem. No peculiarities were observed.

**Action:** Added: "The usual diagnostics were applied and in none of the cases any peculiarities were detected. This provides evidence that the system of equations solved has an unambiguous solution."

**Comment: Lines 137-139: Even with the reduced vertical scale plots, it is again very difficult to see and interpret these results from vectors. The streamfunction should be plotted, as well as difference plots between the initial and final streamfunction.**

**Reply:** As stated above, we will make the data available. Every interested reader can then plot the data in their preferred way. We do not understand what the "difference plots between the initial and final streamfunction" is meant to be. What is the "initial streamfunction"? For the inversion we do not use any prior assumption on the velocity field. Our initial guess is all zero. We do this to be sure that all structure we see comes from the data and not from any prior assumption mapped onto the result.

**Comment: Lines 142-143: Again, I do not easily see the "stratospheric branches of the BDC". Please plot streamfunctions and define what you mean be "branches" (I do understand what is meant, but many readers may not).**

**Reply:** For the representation, we will provide the original data to allow each reader to plot them in their preferred representation.

**Action:** To avoid quibbling about words we have replaced "stratospheric branches of the BDC" with "poleward transport in the SH subtropics at 25 km altitude and in the NH subpolar region at 15 km altitude."

**Comment: Lines 147-148: I'm not sure I agree that the "the slow circulation patterns in the tropopause region and the lower stratosphere are well recovered". If plotted as percent differences, I think**

**some very large discrepancies would emerge.**

**Reply:** And if the true value was zero, even the best recovered velocity would have an infinite error... Percentage values can be very misleading when the reference values are small.

**Comment: Lines 153-154: The right panels of Figure 5 are the only figures in the manuscript that seem to resemble the canonical stratospheric meridional circulation. They show rapid poleward transport by the shallow branch (below 15 km here) and strong tropical upwelling, poleward transport, and high latitude downwelling. No other plot shows a coherent upwelling region like this one does. Of course differences are to be expected given the seasonality of the circulation, but the upwelling branch moves back and forth across the year rather than disappearing.**

**Reply:** We are happy to hear from the reviewer that the panels on the right of Fig. 5 satisfy the picture they are used to. The top right panel of Fig. 5 is just a zoomed version of the top right panel of Fig. 3. It is a result of AN-CISTRUS, restricted to the altitude range the BDC is usually looked at, with a velocity scale that is adjusted to the low velocities appearing here (in contrast to the high velocities that dominate the upper stratosphere and mesosphere). The tropical upwelling is an extremely slow process and is easily masked by the seasonality. It can be seen in the third panel of Figure 2 that we do have the tropical upwelling as a background signal. In the individual months, this signal is, however, superimposed by other processes related to, e.g., seasonality or inter-annual variation (QBO, ENSO, ...).

**Comment: Lines 157-158: While it is true that the second retrieval did not create significantly different patterns than the first, it has not been established that the patterns retrieved in the first place are not artificial given that the algorithm does not appear to have ever been tested with a realistic circulation pattern and realistic tracer distribution.**

**Reply:** What 'realistic' velocity fields do we have available? Models? Funke et al. (2011, their Figs. 14 and 17) have fed 10 different models with the same measured distribution of $NO_y$, which can be considered as an inert tracer on the relevant time-scale. Already after a couple of days, 10 very different distributions were predicted, and the differences were attributed to transport modelling. If model fields are realistic, which of these realities is the real reality? Or do we have parallel universes?

And does the reviewer intend to label our tracer distributions as unrealistic? These are based on MIPAS measurements, and a lot of validation studies have provided evidence of their reliability. What more realistic global tracer distributions do we have available? And even if, as suspected by the reviewer, the

results of the first retrieval were unrealistic, this does not in any way make a model recovery test invalid. We demonstrate that, if we feed ANCISTRUS with trace gas distributions associated with two times, we get back the underlying velocity field. This is exactly the purpose of a model recovery test.

**Comment: Figure 3: There are obviously large differences in the velocities at 60S, 60 km for Feb-Mar. Why don't these show up in the difference plots? There are also other examples where the difference plots do not seem to reflect the visual differences between the top and middle plots.**

**Reply:** At 60S, 60 km for Feb-Mar, the reference field shows values slightly larger than 8; the retrieved field shows values around 6 to 7, and the difference field shown values around 2. We do not see what the problem is. We have randomly checked other instances and did not find any inconsistency either. We do not know what the reviewer is speaking about. Is this unfounded accusation just another rhetoric trick to undermine the credibility of the authors?

**Action:** When checking the figures, we have detected some minor inconsistencies with respect to the layout. These have been removed. These were, however, not related to the comment of the reviewer.

**Comment: Lines 201-204: Why does withholding CFC11 give the opposite signal to CFC12 in the Arctic? If the sinks are properly accounted for, the effective transport for these two species should be similar.**

**Reply:** These gases have their strongest vertical gradients in different altitudes and have quite different lifetimes. Furthermore, we solve an overdetermined system of equations, and measurements are not always perfectly consistent. One gas may try to push the solution into one direction, and the other gas in the opposite one, and the least squares solution is a compromise. Removing one species in some way slightly tips the balance. It should be noted that these differences are quite small compared to the effective velocities (note the factor $10^{-3}$ in the titles of the panels).

**Comment: Line 208-211: I do not understand what is meant by "compressed colour scale". Again, the streamfunction and percent differences might be more useful for seeing the stratospheric changes.**

**Reply:** This means that a larger range of values is covered by the colours.

**Action:** Reworded: "[...] due to the large range of values represented by the colour scales of the figures".

**Comment: Lines 211-212: The water vapor "tape recorder" has been**

**extensively used for deriving vertical transport in the tropics, yet water seems to do nothing to inform the tropical upwelling. Can this be explained?**

**Reply:** Yes, it can be explained. The reason is that the other species include so much information already that adding consistent information from water vapour does not change a lot. This just means that the information from water vapour and that of the other species is pretty much consistent. By the way: as said above, the ascent rate of the tape recorder is exactly the same concept as our pattern displacement concept discussed in Section 2. Does this mean that the ascent rate of the tape recorder reflects only anomalies of the updraft?

**Comment: Lines 232-233: If this is meant to refer to circulation patterns and structures, then I have to say I strongly disagree that there is evidence that ANCISTRUS is fit for purpose. It does indeed generate a consistent set of patterns and structures from a given set of trace gas fields, but there is no evidence that these patterns and structures are physically meaningful in any way. Until this is demonstrated using a known, realistic circulation field with the MIPAS tracer measurements, I cannot recommend publication of this manuscript.**

**Reply:** Model recovery tests as we have performed them are the standard procedure to test inverse schemes. It is the fundamental logic of model recovery tests that some input is generated using some 'surrogate truth' with the forward model and to test if the model is able to reproduce the 'surrogate truth'. Model recovery tests alone do not demonstrate that the structures are physically meaningful, but complemented with the forward model tests in vCG16 they do. The model recovery tests demonstrate that the inversion procedure does what it is supposed to do. All the physics (which makes the results 'physically meaningful') is in the forward model which has been tested independently.

**Comment: Minor comments: Line 1: The wording "allows to infer" is not grammatically correct (it needs a subject). I suggest "provides an inference of".**

**Reply:** Gramatically, 'The direct inversion' is the subject of this sentence.

**Action:** changed to: "allows for the inference of"

**Comment: Line 2: The phrase "both given by" should be "given by both"**

**Reply:** agreed.

**Action:** Changed as suggested.

**Comment: Line 4: Using "have shown" in the past tense makes it sound as if these tests were performed in another paper rather than here.**

**Reply:** agreed.

**Action:** changed to "show".

**Comment: Abstract in general: The abstract does not provide sufficient context for this work or provide any indication of the meaningfulness of the results.**

**Reply:** The context is given in the first two sentences.

**Action:** We have added: "With these tests the reliability of the method has been established."

**Comment: Lines 66-67: The phrase "does effectively not work" should be "effectively does not work"**

**Reply:** agreed.

**Action:** This part had already been reworded in reply to other comments.

**Comment: Line 84: Should "or equatorward transport" be "of equatorward transport"?**

**Reply:** agreed.

**Action:** corrected.

**Comment: Line 88: The reference should be made to "bottom panels" rather than "right panels". Line 140: I believe "September October 2005" should be "March April 2005"**

**Reply:** agreed.

**Action:** corrected.

**Summary Reply to Review #2:** This is a technical paper which presents tests of the ANCISTRUS analysis tool. The review is dominated by a dispraisal of the trace gas and velocity distributions we work with. These, however, are not the topic of this paper. The topics of this paper are:

  1. Which are the dominating information pathways explored by ANCISTRUS?

2. Can ANCISTRUS reproduce reference fields when it is fed with trace gas distributions consistent with these fields?

3. To which degree do ANCISTRUS results depend on the regularization chosen?

4. Which is the information content provided by different trace gases?

For the few comments which are related to these key questions we think to have shown that these are based on a fundamental misunderstanding of the purpose and the rationale of the related tests. Many of the comments do not discuss these tests at all and are thus not relevant to this validation paper. The only criticism directly related to this manuscript is the choice of the velocity fields used for the model recovery test. The reviewer does not accept that we use velocity fields generated with ANCISTRUS, however, no conclusive argument is present about what is wrong with this approach. The dismissive and false statement "all that is shown is that ANCISTRUS can retrieve the same field twice" does not refute the logic of our tests at any rate. These model recovery tests are a necessary precondition for any meaningful comparison of ANCISTRUS results with data from chemistry-climate or chemistry-transport models. The reviewer suggests instead to use fields from a climate model or a CTM for the ANCISTRUS model recovery tests. On the face of it, this suggestion sounds plausible, but we have presented arguments why this is not adequate. There are three options how such a model-based test could be organized:

1. The simplest approach would be to directly compare model velocity fields (transformed to 2D) to ANCISTRUS fields of the same time period. This test would not be a model recovery test and would fully rely on models representing the truth. This is, however, by no means guaranteed and thus this approach is not adequate for the validation of ANCISTRUS.

2. One could use modeled VMR fields, feed them in ANCISTRUS, and compare the ANCISTRUS velocity fields with those from the model (transformed to 2D). Logically, this test would be flawless, but there are practical issues which rule out this test: There are not so many models which provide VMR-distributions of all the species ANCISTRUS needs. The few models we have seen so far which provide these were not useful for this purpose, for two reasons: (a) The mixing ratios, particularly in the upper part of the ANCISTRUS domain, deviated much from the MIPAS measurements. For some species, these VMRs were considerably lower than the MIPAS measurements and thus contained no sizeable amount of information at certain altitudes where ANCISTRUS needs this information. (b) In the upper part of the ANCISTRUS domain, the modeled VMR fields were much smoother than those measured by MIPAS. The VMR structures which are transported contain a large amount of the information exploited by ANCISTRUS but they were not present in the model data we have seen so far. We consider it as inadequate to test ANCISTRUS with test data which do not contain the information needed, because from

these results we do not learn how ANCISTRUS behaves when fed with real measurement VMR data.

3. One could also use only the 2D velocity fields from the model, apply them to MIPAS VMR fields at time $t_1$ and calculate VMR fields for time $t_2$. Both VMR fields are then fed into ANCISTRUS, and the resulting velocities are compared to the 2D model velocities. This was actually the suggestion by reviewer #2. This approach, however, will fail because the information on the velocities is not only in the difference $VMR(t_2)$ minus $VMR(t_1)$, but also in the absolute VMR values, due to the advection-sink relation. We have shown this in Section 2 of the paper. Thus, since the VMRs and the velocities would not be consistent in this test scenario, we cannot expect ANCISTRUS to retrieve the correct velocities.

With this, we think we have refuted the only substantial criticism of review #2.

Thomas von Clarmann, Udo Grabowski, Gabriele P. Stiller, Beatriz M. Monge-
Sanz, Norbert Glatthor, and Sylvia Kellmann, An observation-based climatol-

[revised manuscript text omitted]

---

## Referee Report (RR1)

I am extremely disappointed by the argumentative and imperious nature of the authors' response to my review, which feels very much like an attempt to intimidate me as a reviewer and is detrimental to the entire peer review process. Every comment that I made in my initial review was interpreted by the authors in the very worst possible light when in fact I meant almost none of what they inferred from my comments. While I had what I believe to be legitimate scientific concerns over the content of the paper, I bore no ill will toward the authors nor did I mean for any of my comments to be personally directed at them. In fact, I was so interested in the concept of the paper that I accepted the reviewer position despite numerous personal challenges I am facing as a result of the COVID-19 pandemic. The authors have shown quite clearly, however, that they are not willing to entertain constructive criticism of their work and that they are uninterested in engaging seriously in the peer review process. While I have completed a second review in the interest of fulfilling my obligation as a reviewer, I will no longer accept review requests for manuscripts written by these authors.

Jessica L. Neu
Principal Scientist
Jet Propulsion Laboratory, California
Institute of Technology
Jesscia.L.Neu@jpl.nasa.gov
+1 (818) 354 0773

Given the clarifications added in the revised version of the manuscript, I recommend publication in ACP but would appreciate the following comments being addressed:

The authors refer to the ANCISTRUS-derived circulation as the "Brewer Dobson Circulation" in several instances throughout the manuscript. It would be very helpful if the authors could provide an explanation of the relationship between the 2-D circulation described by ANCISTRUS and the 2-D residual circulation in the Transformed Eulerian Mean framework, which is often used interchangeably with the term "Brewer Dobson Circulation".

Lines 135-136: I still disagree that the upper left panels of Figures 1 and 2 necessarily represent a "realistic circulation field" and believe it would be more accurate to say that both information on transported structures and chemical sinks provide circulation-relevant information to the inversion system.

Lines 215-217: Typically, when people refer to the "branches" of the Brewer Dobson Circulation, they are referring to a "lower branch" at ~100-70 hPa and an "upper branch" at ~50-10 hPa. Here the authors are referring to a Northern Hemisphere branch and Southern Hemisphere branch, and the usage is confusing. Perhaps "cells" would be a better description than "branches".

Lines 337-338: The authors state that the main application of ANCISTRUS is to investigate the structure and possible changes of the Brewer Dobson Circulation, but it is more accurate to say that the method can investigate the structure and possible changes of the circulation as derived from ANCISTRUS until such time as a clear understanding of the relationship between the ANCISTRUS-derived circulation and either the diabatic circulation or the residual circulation is established.

---

## Author Response (AR2)

Editor comment to the author:

Comment: Thank you for your efforts in revising the manuscript. You will see that the referees are mainly satisfied with the revisions but you will need to address the remaining points raised by reviewer 2 before the manuscript can be accepted for publication in ACP. Furthermore, this manuscript is title as "Part 2" to another study submitted in ACPD for which the review process is currently stopped. I cannot accept this manuscript as a "Part 2" to a study where "Part 1" is not accepted. The authors should revise the title of the manuscript accordingly.

**Reply:** The "Part 1" paper we refer to was already published in 2016: "Direct inversion of circulation and mixing from tracer measurements - Part 1: Method", T. von Clarmann and U. Grabowski, Atmos. Chem. Phys., 16, 14563-14584, 2016, doi:10.5194/acp-16-14563-2016. The long delay between these papers is explained by the fact that this research topic is not our primary job task. Thus the title with "part 2" does not depend on the acceptance of the ACPD paper still in review, "An observation-based climatology of middle atmospheric meridional circulation", T. von Clarmann, U. Grabowski, G. Stiller, B. M. Monge-Sanz, N. Glatthor and S. Kellmann, Atmos. Chem. Phys. Discuss., https://doi.org/10.5194/acp-2019-704, 2019.

Action: Since we think that your concern is based on a misunderstanding, we have not changed the title of the paper. We have, however, doublechecked all the references to these papers to avoid any possible confusion.

**Report #2 by Anonymous Referee #1:**

**Reply:** Lots of thanks for the appreciation of our work!

**Report #1 by Jessica Neu:**

Comment: I am extremely disappointed by the argumentative and imperious nature of the authors' response to my review, which feels very much like an attempt to intimidate me as a reviewer and is detrimental to the entire peer review process. Every comment that I made in my initial review was interpreted by the authors in the very worst possible light when in fact I meant almost none of what they inferred from my comments. While I had what I believe to be legitimate scientific concerns over the content of the paper, I bore no ill will toward the authors nor did I mean for any of my comments to be personally directed at them. In fact, I was so interested in the concept of the paper that I accepted the reviewer position despite numerous personal challenges I am facing as a result of the COVID-19 pandemic. The authors have shown quite clearly, however, that they are not willing to entertain constructive criticism of their work and that they are uninterested in engaging seriously in the peer review process. While I have completed a second review in the interest of fulfilling my obligation as a reviewer, I will no longer accept review requests for manuscripts written by these authors.

**Reply:** We do not want to comment on this.

Action: None.

Comment: Given the clarifications added in the revised version of the manuscript, I recommend publication in ACP but would appreciate the following comments being addressed: The authors refer to the ANCISTRUS-derived circulation as the "Brewer Dobson Circulation" in several instances throughout the manuscript. It would be very helpful if the authors could provide an explanation of the relationship between the 2-D circulation described by ANCISTRUS and the 2-D residual circulation in the Transformed Eulerian Mean framework, which is often used interchangeably with the term "Brewer Dobson Circulation".

**Reply:** To our understanding, the Brewer-Dobson-Circulation includes the transport caused by zonal mean gradients and velocities, transport caused by the correlations between the perturbations of mixing ratios and velocities (eddy contributions), and physical mixing. While the 2-D residual circulation in the Transformed Eulerian Mean framework covers only the first two mechanisms, ANCISTRUS-derived effective velocities currently include also the effect of physical mixing, as the age of air concept does.

Action: We have added at the end of the first paragraph of the introduction: "Beyond this, the ANCISTRUS-derived effective velocities also include a contribution by physical mixing and thus are not directly comparable to the 2-D residual circulation in the Transformed Eulerian Mean framework."

Comment: Lines 135-136: I still disagree that the upper left panels of Figures 1 and 2 necessarily represent a "realistic circulation field" and believe it would be more accurate to say that both information on transported structures and chemical sinks provide circulationrelevant information to the inversion system.

**Reply:** In the figure captions we make no claims about a 'realistic circulation field' but we speak about 'realistic assumptions'. We still think that it is realistic to assume that there are sinks in the atmosphere and that the atmosphere is not stationary. The only instance where we use the term 'realistic circulation field' is the statement 'In summary, it is evident that both sources of information have to be exploited to infer a realistic circulation field.' This is

to be understood as a necessary, not a sufficient condition. That is to say, this statement does not imply any claim that our circulation fields are necessarily realistic. To avoid misunderstanding, we have changed the wording.

Action: Old: "In summary, it is evident that both sources of information have to be exploited to infer a realistic circulation field."

New: "In summary, it is evident that both sources of information contribute to the resulting circulation field, and it is necessary to exploit both of them to infer a realistic circulation field."

Comment: Lines 215-217: Typically, when people refer to the "branches" of the Brewer Dobson Circulation, they are referring to a "lower branch" at 100-70 hPa and an "upper branch" at 50-10 hPa. Here the authors are referring to a Northern Hemisphere branch and Southern Hemisphere branch, and the usage is confusing. Perhaps "cells" would be a better description than "branches".

**Reply:** We do agree that the use of the term 'branch' in a generic sense, without reference to the technical terms 'upper branch' and lower branch' may be confusing. The suggested term 'cell', however, makes us associate a closed circulation cell, but the criticism refers to instances where we refer only to a part of the circulation, e.g., 'upwelling circulation branch', 'subsiding branch'. Thus we think that 'cell' is not the optimal substitute for the generic term 'branch'. Instead, we now refer to 'segments' of the circulation whereever the deep or shallow branch of the circulation are not meant.

Action: Reworded, using 'segment' in most instances.

Comment: Lines 337-338: The authors state that the main application of ANCISTRUS is to investigate the structure and possible changes of the Brewer Dobson Circulation, but it is more accurate to say that the method can investigate the structure and possible changes of the circulation as derived from ANCISTRUS until such time as a clear understanding of the relationship between the ANCISTRUS-derived circulation and either the diabatic circulation or the residual circulation is established.

**Reply:** This sentence was inserted following a suggestion of the other reviewer, thus we are reluctant to change it back. Parts of the community understand the BDC as residual circulation plus mixing (see, e.g., recent work by Garny and colleagues). The BDC obtained its name because it explains the distributions of water vapour (Brewer) and ozone (Dobson) in the stratosphere. It is thus an observation-based, not a theoretical term. The theory of the residual circulation, and even more the theory of the diabatic circulation are based on idealizing assumptions and, due to their disregard of mixing, explain only a part of the whole phenomenon. Thus we find it challengeable to equate the BDC with either of these theoretical frameworks and we thus think that there is nothing wrong with the statement that the main application of ANCISTRUS is to investigate the structure and possible changes of the Brewer Dobson Circulation. All justification to use the age of air as a diagnostic of the BDC applies equally to our circulation fields, which can be regarded as the inverse age different per path element.

Action: As stated above, we have added at the end of the first paragraph of the introduction: "Beyond this, the ANCISTRUS-derived effective velocities also include 
[revised manuscript text omitted]